# Sequential Group Composition: A Window into the Mechanics of Deep Learning

Giovanni Luca Marchetti [* 1]   Daniel Kunin [* 2]   Adele Myers [3]   Francisco Acosta [3]   Nina Miolane [3]

## Abstract

How do neural networks trained over sequences acquire the ability to perform structured operations, such as arithmetic, geometric, and algorithmic computation? To gain insight into this question, we introduce the *sequential group composition* task. In this task, networks receive a sequence of elements from a finite group encoded in a real vector space and must predict their cumulative product. This task can be order-sensitive and cannot be solved by a linear model. Our analysis isolates the roles of the group structure, encoding statistics, and sequence length in shaping learning. We prove that two-layer networks from vanishing initialization learn this task one irreducible representation of the group at a time in an order determined by the Fourier statistics of the encoding. To perfectly learn the task, these networks require a hidden width exponential in the sequence length $k$. In contrast, we construct deeper architectures that exploit associativity to dramatically improve this scaling: recurrent neural networks can compose elements sequentially in $k$ steps, while multilayer networks can compose adjacent pairs in parallel in $\log k$ layers. Overall, the sequential group composition task offers a tractable window into the mechanics of deep learning.

## 1. Introduction

Natural data is full of symmetry: reindexing the atoms of a molecule leaves its physical properties unchanged; translating or reflecting an image preserves the scene; and re-ordering words sometimes preserves semantic meaning and sometimes does not—revealing both commutative and non-commutative structure. Consequently, many tasks we train neural networks on are, at their core, computations over

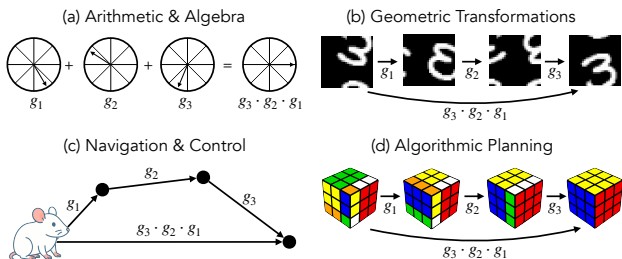

*Figure 1.* **A unifying abstraction.** Across arithmetic, perception, navigation, and planning, many sequence tasks require learning to compose transformations from examples. Motivated by this shared structure, we introduce the *sequential group composition* task—a unifying abstraction where networks learn to map a sequence of group elements to their cumulative product (1).

groups that require learning to *compose* transformations rather than merely recognize them. Yet, it remains unclear *how* standard architectures acquire and represent these composition rules—what features do they learn and in what order. This paper addresses that gap by developing an analytic account of how simple networks learn to compose elements of finite groups represented in a real vector space.

In this paper, we analyze how neural networks learn group composition through gradient-based training on sequences. Given any finite group $G$, Abelian or non-Abelian, the neural network is tasked with learning a map from a sequence of group elements to their ordered (from left to right) cumulative product:

$$(g_1, \ldots, g_k) \in G^k \mapsto \prod_{i=1}^{k} g_i \in G. \tag{1}$$

Although idealized, this setting is quite general and captures the essence of many natural problems (see Figure 1). Solving puzzles such as the Rubik's Cube amounts to composing a sequence of moves, each a group element. Tracking the trajectory of a body through physical space requires composing rigid motions or integrating successive displacements. Beyond puzzles and physics, groups also underpin information processing and algorithm design, where complex computations arise from composing simple operations. A canonical example is modular addition—computing sums of integers modulo $p$—which corresponds to the binary case $k = 2$ over the cyclic group $C_p$.

---
[*]Equal contribution  [1]KTH Royal Institute of Technology [2]UC Berkeley [3]UC Santa Barbara. Correspondence to: Giovanni Luca Marchetti <glma@kth.se>, Daniel Kunin <kunin@berkeley.edu>.

*Proceedings of the 43rd International Conference on Machine Learning*, Seoul, South Korea. PMLR 306, 2026. Copyright 2026 by the author(s).

We cast the group composition task as a regression problem: a neural network $f : \mathbb{R}^{k|G|} \to \mathbb{R}^{|G|}$ receives as input $k$ group elements, $g_1, \ldots, g_k$, and is trained to estimate their product $g_1 \cdots g_k$. The group elements are encoded with a fixed *encoding vector* $x \in \mathbb{R}^{|G|}$, which we discuss in Section 3.1. This formulation highlights a central challenge: the number of possible input sequences grows exponentially with $k$. While memorization is possible in principle for fixed $k$ and $|G|$, any solution that scales efficiently with sequence length requires the network to uncover and represent the algebraic structure of the group. Our analysis and experiments show that networks do so by progressively decomposing the task into the irreducible representations of the group, learning these components in a greedy order based on the encoding vector $x$. Different architectures realize this process in distinct ways: while two-layer networks attempt to compose all $k$ elements at once, requiring width on the order of $2^k$, recurrent models can build products sequentially in $k$ steps, and multilayer networks can combine elements in parallel in $\log k$ layers. Our results reveal both a *universality* in the dynamics of feature learning and a diversity in the *efficiency* with which different architectures exploit the associativity of the task.

**Our contributions.** In this work we introduce the *sequential group composition* task and prove that it admits several properties that make it especially well suited for studying how neural networks learn from sequences:

1. **Order sensitive and nonlinear (Section 3).** We establish that the task, which depending on the group may be order-sensitive or order-insensitive, cannot be solved by a (deep) linear network, as it requires nonlinear interactions between inputs.

2. **Tractable feature learning (Section 4).** We show that the task admits a group-specific Fourier decomposition, enabling a precise analysis of learning for a class of two-layer networks. In particular, we prove how the group Fourier statistics of the encoding vector $x$ determine what features are learned and in what order.

3. **Compositional efficiency with depth (Section 5).** We demonstrate that while two-layer networks require a hidden width exponential in the sequence length $k$, deep networks can identify efficient solutions by exploiting associativity to compose intermediate representations.

Overall, these results position sequential group composition as a principled lens for developing a mathematical theory of how neural networks learn from sequential data, with broader implications, limitations, and next steps discussed in Section 6.

## 2. Related Work

Our work engages with three fields: *mechanistic interpretability*, where we identify the Fourier features used for group composition; *learning dynamics*, where we explain how these features emerge through stepwise phases of training; and *computational expressivity*, where we characterize how these phases scale with sequence length depending on architectural bias toward sequential or parallel computation.

**Mechanistic interpretability.** A large body of recent work has sought to reverse-engineer trained neural networks to identify the algorithms they learn to implement (Olah et al., 2020; Elhage et al., 2021; Olsson et al., 2022; Elhage et al., 2022; Bereska & Gavves, 2024; Sharkey et al., 2025). A common strategy in this literature is to analyze simplified tasks that reveal how networks represent computation at the level of weights and neurons. Among the most influential case studies are networks trained to perform modular addition (Power et al., 2022). It has been shown by numerous empirical studies that networks trained on this task develop internal *Fourier features* and exploit trigonometric identities to implement addition as rotations on the circle (Nanda et al., 2023; Gromov, 2023; Zhong et al., 2024). Related Fourier features have also been observed in networks trained on binary group composition tasks (Chughtai et al., 2023; Stander et al., 2023; Morwani et al., 2023; Tian, 2024; Li et al., 2024; Wu et al., 2025; McCracken et al., 2026) and in large pre-trained language models performing arithmetic (Zhou et al., 2024; Kantamneni & Tegmark, 2025; Moisescu-Pareja et al., 2025). Several works have sought to explain why such structure emerges, linking it to the task symmetry (Marchetti et al., 2024), simplicity biases of gradient descent (Morwani et al., 2023; Tian, 2024), and most recently a framework for feature learning in two-layer networks (Kunin et al., 2025; He et al., 2026). Our work extends these insights to group composition over sequences, and rather than inferring circuits solely from empirical inspection, we derive from first principles how networks progressively acquire these Fourier features through training. We collect technical remarks on how our work relates to previous group-theoretical constructions in Appendix E.

**Learning dynamics.** A complementary line of research investigates how computational structure emerges during training by analyzing the trajectory of gradient descent rather than the final trained model. A consistent empirical finding is that networks acquire simple functions first, with more complex features appearing only later in training (Arpit et al., 2017; Kalimeris et al., 2019; Barak et al., 2022). This staged progression—sometimes described as *stepwise* or *saddle-to-saddle*—is marked by extended plateaus in the loss punctuated by sharp drops (Jacot et al., 2021). These dynamics have been theoretically characterized across a range

of simple settings (Gidel et al., 2019; Li et al., 2020; Pesme & Flammarion, 2023; Zhang et al., 2025b;a). Of particular relevance is the *Alternating Gradient Flow (AGF)* framework recently introduced by Kunin et al. (2025), which unifies many such analyses and explains the stepwise emergence of Fourier features in modular addition. Building on this perspective, we show that networks trained on the sequential group composition task acquire Fourier features of the group in a greedy order determined by their importance.

**Computational expressivity.** Algebraic and algorithmic tasks have also become canonical testbeds for probing the computational expressivity of neural architectures (Liu et al., 2022; Barkeshli et al., 2026). Classical results established that sufficiently wide two-layer networks can approximate arbitrary functions, yet the ability to (efficiently) find these solutions depends on the architecture. Recent analyses have examined the dominance of transformers in sequence modeling, contrasting their performance with that of RNNs and feedforward MLPs. Across these works, a consistent picture emerges: transformers efficiently implement compositional algorithms with logarithmic depth by exploiting parallelism, while recurrent models realize the same computations sequentially with linear depth, and shallow networks require exponential width (Liu et al., 2022; Sanford et al., 2023; 2024a;b; Bhattamishra et al., 2024; Jelassi et al., 2024; Wang et al., 2025; Mousavi-Hosseini et al., 2025). Our analysis confirms this lesson in the context of group composition, enabling a precise characterization of how the architecture determines not only what can be computed, but also how efficiently such computations are learned.

## 3. A Sequence Task with Structure & Statistics

In this section, we begin by reviewing mathematical background on groups and harmonic analysis over them, which will be used throughout the paper. We then formalize the *sequential group composition* task and highlight the properties that make it particularly well suited for analysis.

### 3.1. Brief Primer on Harmonic Analysis over Groups

**Groups.** *Groups* formalize the idea of a set of (invertible) transformations or symmetries, that can be composed.

**Definition 3.1.** A *group* is a set $G$ equipped with a binary operation $G \times G \to G$ denoted by $(g, h) \mapsto gh$, with an *inverse element* $g^{-1} \in G$ for each $g \in G$ and an *identity element* $1 \in G$ such that for all $g, h, k \in G$:

| Associativity | Inversion | Identity |
| --- | --- | --- |
| $g(hk) = (gh)k$ | $g^{-1}g = gg^{-1} = 1$ | $g1 = 1g = g$ |

A group is *Abelian* if its elements commute ($gh = hg$ for all $g, h \in G$); otherwise it is *non-Abelian*. Abelian groups model order-insensitive transformations, such as the *cyclic*

group $C_p = \mathbb{Z}/p\mathbb{Z}$, which consists of integers modulo $p$ with addition modulo $p$ as the group operation. Non-Abelian groups capture order-sensitive transformations, such as the *dihedral group* $D_p$, which consists of all rotations and reflections of a regular $p$-gon. Here the order matters, since rotating then reflecting does not yield the same result as reflecting then rotating, as shown in Figure 2a for $D_3$.

**Group representations.** Elements of any group can be *represented* concretely as invertible matrices, where composition corresponds to matrix multiplication. This allows group operations to be analyzed through linear algebra. We focus on representations with $n$-dimensional *unitary*[1] matrices, which form the unitary group $U(n) = \{A \in \mathbb{C}^{n \times n} \mid A^\dagger A = I\}$, where $\dagger$ denotes the conjugate transpose.

**Definition 3.2.** An $n$-dimensional *unitary representation* of $G$ is a map $\rho: G \to U(n)$ such that $\rho(gh) = \rho(g)\rho(h)$ for all $g, h \in G$, i.e., a homomorphism between $G$ and $U(n)$.

An important representation for a finite group $G$ is the (left) *regular representation*, which maps each element $g \in G$ to a $|G| \times |G|$ permutation matrix $\lambda(g)$ that acts on the vector space $\mathbb{C}^{|G|}$ generated by the one-hot basis $\{\mathbf{e}_h : h \in G\}$:

$$\lambda(g)\, \mathbf{e}_h = \mathbf{e}_{gh}, \qquad h \in G. \tag{2}$$

A vector in $\mathbb{C}^{|G|}$ can be thought as a complex-valued signal over $G$, whose coordinates get permuted by $\lambda(g)$ according to the group composition; see Figure 2b.

The regular representation, which has dimension equal to the *order* of the group $|G|$, can be decomposed into lower-dimensional unitary representations that still faithfully capture the group's structure. These representations, which cannot be broken down any further, are called *irreducible representations* (or *irreps*) and serve as the fundamental building blocks of every other unitary representation. For a finite group $G$, there exists a finite number of irreps up to isomorphism. For Abelian groups, the irreps are one-dimensional, while non-Abelian groups necessarily include higher-dimensional irreps that capture their order-sensitive structure. Every group has a one-dimensional *trivial* irrep, denoted $\rho_{\text{triv}}$, which maps each $g \in G$ to the scalar 1. Let $\mathcal{I}(G)$ denote the set of irreps up to isomorphism, and $n_\rho$ the dimension of $\rho \in \mathcal{I}(G)$. See Figure 2b for an illustration of the regular and irreducible representations of $D_3$.

**Orbit-based encoding of $G$.** Representation theory translates group structure into unitary matrices, but to train neural networks we require a real-valued encoding $G \to \mathbb{R}^{|G|}$ that reflects the group structure. We obtain such an encoding by taking the *orbit* of a fixed *encoding vector* $x \in \mathbb{R}^{|G|}$ under the regular representation: $g \mapsto \lambda(g)^\mathsf{T}x$. For $x = e_1$,

---

[1]This is not restrictive, since any group representation can converted into a unitary one via an appropriate Hermitian metric.

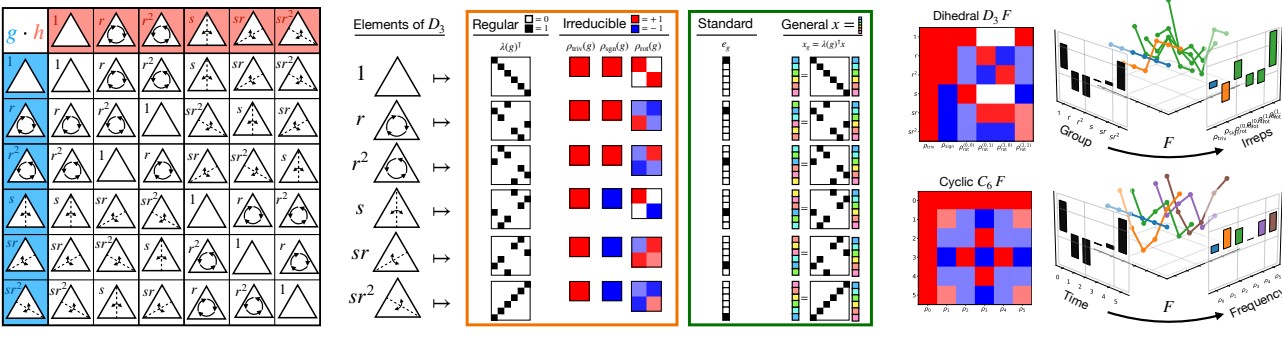

*(a)* Dihedral Group $D_3$      *(b)* Representations and orbit-based encodings of $D_3$      *(c)* Fourier transform of $D_3$ and $C_6$

*Figure 2.* **Visual introduction to abstract harmonic analysis.** (a) The dihedral group $D_3$ consists of all rotations and reflections of a regular triangle, a canonical non-Abelian group where composition is order-dependent. (b) Its regular representation acts on $\mathbb{C}^{|G|}$ as $6 \times 6$ permutation matrices, which decompose into two one-dimensional and one two-dimensional irreducible representations (irreps). We encode $G$ by taking the orbit of a fixed encoding vector $x \in \mathbb{R}^6$ under the regular representation; this reduces to the standard one-hot encoding when $x = e_1$. (c) The Fourier transform is a unitary change of basis built from the irreps of $G$: see, e.g., how its first row corresponds to flattening the irreps of the identity element 1. It decomposes a signal $x \in \mathbb{R}^{|G|}$ into its irrep components, with coefficients $\widehat{x} = F^\dagger x$. This construction generalizes the classical DFT, recovered when $G = C_p$. Here we show the Fourier transform for $D_3$ and $C_6$.

this reduces to the standard one-hot encoding $g \mapsto e_g$. For convenience we denote $x_g = \lambda(g)^\mathsf{T} x$. For general $x$, the orbit $\{x_g\}_{g \in G}$ depends on both the structure of the group $G$ and the statistics of the encoding vector $x$. Figure 2b illustrates this encoding for $D_3$.

**Group Fourier transform.** The decomposition of the regular representations into the irreducible representations is achieved by a change of basis $F \in \mathbb{C}^{|G| \times |G|}$ that simultaneously block-diagonalizes $\lambda(g)$ for all $g \in G$. This change of basis is the *group Fourier transform*.

**Definition 3.3.** The *Fourier transform* over a finite group $G$ is the map $\mathbb{C}^{|G|} \to \bigoplus_{\rho \in \mathcal{I}(G)} \mathbb{C}^{n_\rho \times n_\rho}$, $x \mapsto \widehat{x}$, defined as:

$$\widehat{x}[\rho] = \sum_{g \in G} \rho(g)^\dagger x[g] \quad \in \mathbb{C}^{n_\rho \times n_\rho}, \qquad (3)$$

where $x[g] \in \mathbb{C}$ indexes the $g$ element of $x$. Flattening all blocks $\widehat{x}[\rho]$ yields a vector $\widehat{x} = F^\dagger x$.

Definition 3.3 generalizes the classical discrete Fourier transform (DFT). To see this, consider the cyclic group $C_p$. The irreps of $C_p$ are one-dimensional and correspond to the $p$ roots of unity, $\rho_k(g) = e^{2\pi \mathrm{i} g k / p}$ for $k \in \{0, \ldots, p-1\}$, where $\mathrm{i} = \sqrt{-1}$ is the imaginary unit. Substituting these irreps into Definition 3.3 yields exactly the standard DFT, and the change-of-basis matrix $F$ coincides with the usual DFT matrix. In this sense, the Fourier transform over a finite group generalizes the classical DFT: the irreps of $G$ act as "matrix-valued harmonics" that extend complex exponentials to non-Abelian settings. See Figure 2c for a depiction of the Fourier transform for $D_3$.

**Harmonic analysis.** Equipped with a Fourier transform, we can extend the familiar tools of classical harmonic analy-

sis beyond the cyclic case to harmonic analysis over groups (Folland, 2016). Importantly, the group Fourier transform satisfies both a *convolution theorem* and a *Plancherel theorem*, see Appendix A for details. To state these results, we introduce a natural inner product and norm on the irrep domain, which we will use throughout our analysis.

**Definition 3.4.** For $\rho \in \mathcal{I}(G)$ and $A, B \in \mathbb{C}^{n_\rho \times n_\rho}$, define the *inner product* $\langle A, B \rangle_\rho := n_\rho \mathrm{Tr}(A^\dagger B)$. The *power of* $x$ at $\rho$ is the induced norm $\|\widehat{x}[\rho]\|_\rho^2 := \langle \widehat{x}[\rho], \widehat{x}[\rho] \rangle_\rho$.

The power generalizes the squared magnitude of a Fourier coefficient in the classical DFT, capturing the energy of the matrix-valued coefficient $\widehat{x}[\rho]$. The $n_\rho$ normalization is chosen such that the Fourier transform is unitary and the total energy decomposes across irreps as $\|x\|^2 = \frac{1}{|G|} \sum_{\rho \in \mathcal{I}(G)} \|\widehat{x}[\rho]\|_\rho^2$, which is the Plancherel theorem.

### 3.2. The Sequential Group Composition Task

The *sequential group composition* task is a regression problem. Given a finite group $G$ and an encoding vector $x \in \mathbb{R}^{|G|}$, a neural network $f$ receives as input a sequence of encoded elements, $x_{\mathbf{g}} := (x_{g_1}, \ldots, x_{g_k}) \in \mathbb{R}^{k|G|}$, and is trained to estimate the encoding of their composition $x_{g_1 \cdots g_k} \in \mathbb{R}^{|G|}$. The network is trained to minimize the mean squared error loss over all sequences of length $k$:

$$\mathcal{L}(\Theta) = \frac{1}{2|G|^k} \sum_{\mathbf{g} \in G^k} \left\| x_{g_1 \cdots g_k} - f(x_{\mathbf{g}}; \Theta) \right\|^2. \qquad (4)$$

The task necessarily requires nonlinear interactions between the inputs:

**Lemma 3.5.** *Let $x$ be a nontrivial ($x \neq 0$) and mean centered ($\widehat{x}[\rho_{\mathrm{triv}}] = \langle x, \mathbf{1} \rangle = 0$) encoding. There is no linear*

map $\mathbb{R}^{k|G|} \to \mathbb{R}^{|G|}$ sending $x_{\mathbf{g}}$ to $x_{g_1 \cdots g_k}$ for all $\mathbf{g} \in G^k$.

See Section A.1 for a proof. Consequently, the simplest standard architecture capable of perfectly solving the task is a two-layer network with a polynomial activation, which we study in the following section.

## 4. Tractable Feature Learning Dynamics

In this section, we consider how a two-layer network learns the sequential group composition task in the vanishing initialization limit. For an input sequence encoded as $x_{\mathbf{g}} \in \mathbb{R}^{k|G|}$, the output computed by the network is:

$$f(x_{\mathbf{g}}; \Theta) = W_{\text{out}} \, \sigma \left( W_{\text{in}} \, x_{\mathbf{g}} \right), \qquad (5)$$

where $W_{\text{in}} \in \mathbb{R}^{H \times k|G|}$ embeds the input sequence into a hidden representation, $\sigma$ is an element-wise polynomial of degree $k$ that is monic (the leading term is $z^k$) and origin-passing ($\sigma(0) = 0$), $W_{\text{out}} \in \mathbb{R}^{|G| \times H}$ unembeds the hidden representation, and $\Theta = (W_{\text{in}}, W_{\text{out}})$. This computation can also be expressed as a sum over the $H$ hidden neurons as $f(x_{\mathbf{g}}; \Theta) = \sum_{i=1}^{H} f(x_{\mathbf{g}}; \theta_i)$, where

$$f(x_{\mathbf{g}}; \theta_i) = w^i \, \sigma \left( \sum_{j=1}^{k} \langle u_j^i, x_{g_j} \rangle \right). \qquad (6)$$

Here, $u^i = (u_1^i, \ldots, u_k^i) \in \mathbb{R}^{k|G|}$ and $w^i \in \mathbb{R}^{|G|}$ denote input and output weights for the $i^{\text{th}}$ neuron, i.e., the $i^{\text{th}}$ row and column of $W_{\text{in}}$ and $W_{\text{out}}$ respectively, and $\theta_i = (u_1^i, \ldots, u_k^i, w^i)$. We study the *vanishing initialization limit*, where the parameters are drawn from a random initialization $\theta_i(0) \sim \mathcal{N}(0, \alpha^2)$ and we take the limit $\alpha \to 0$. The parameters then evolve under a *time-rescaled gradient flow*, $\dot{\theta}_i = -\eta_{\theta_i} \nabla_{\theta_i} \mathcal{L}(\Theta)$, with a neuron-dependent learning rate $\eta_{\theta_i} = \|\theta_i\|^{1-k} \log(1/\alpha)$ (see Kunin et al. (2025) for details), minimizing the mean squared error loss Equation (4).

### 4.1. Alternating Gradient Flows (AGF)

Recent work by Kunin et al. (2025) introduced *Alternating Gradient Flows* (AGF), a framework describing gradient dynamics in two-layer networks under vanishing initialization. Their key observation is that in this regime hidden neurons operate in one of two states—*dormant*, with parameters near the origin ($\|\theta_i\| \approx 0$) that have negligible influence on the output, or *active*, with parameters far from the origin ($\|\theta_i\| \gg 0$) that directly shape the output. Dormant neurons $\mathcal{D} \subseteq [H]$ evolve slowly, independently identifying directions of maximal correlation with the residual. Active neurons $\mathcal{A} \subseteq [H]$ evolve quickly, collectively minimizing the loss and forming the residual. Initially all neurons are dormant; during training, they undergo abrupt activations one neuron at a time. AGF describes these dynamics as an alternating two-step process:

*1. Utility maximization.* Dormant neurons compete to align with informative directions in the data, determining which feature is learned next and when it emerges. Assuming the prediction over the active neurons $f(x_{\mathbf{g}}; \Theta_{\mathcal{A}})$ is stationary, the utility of a dormant neuron is defined as

$$\mathcal{U}(\theta_i) = \frac{1}{|G|^k} \sum_{\mathbf{g} \in G^k} \langle f(x_{\mathbf{g}}; \theta_i), x_{g_{1:k}} - f(x_{\mathbf{g}}; \Theta_{\mathcal{A}}) \rangle, \quad (7)$$

and the corresponding optimization problem is

$$\forall i \in \mathcal{D} \qquad \text{maximize} \quad \mathcal{U}(\theta_i) \quad \text{s.t.} \quad \|\theta_i\| = 1. \qquad (8)$$

Dormant neuron(s) attaining maximal utility will eventually become active (see Kunin et al. (2025) for details).

*2. Cost minimization.* Once active, a neuron rapidly increases in norm, consolidating the learned feature and causing a sharp drop in the loss. In this phase, the parameters of the active neurons $\Theta_{\mathcal{A}}$ collaborate to minimize the loss:

$$\text{minimize} \quad \mathcal{L}(\Theta_{\mathcal{A}}) \qquad (9)$$

Iterating these two phases produces the characteristic staircase-like loss curves of small-initialization training, where plateaus correspond to utility maximization and drops to cost minimization.

### 4.2. Learning Group Composition with AGF

We now apply the AGF framework to characterize how a two-layer MLP with polynomial activation learns group composition. Our analysis reveals a step-wise process, where irreps of $G$ are learned in an order determined by the Fourier statistics of $x$, as shown in Figure 3. During utility maximization, neurons specialize, independently, to the real part of a single irrep. During cost minimization, we assume $N$ neurons have simultaneously activated aligned to the same irrep, and remain aligned while jointly minimizing the loss. Within these irrep-constrained subspaces, we can solve the loss minimization problem, revealing the function learned by each group of aligned neurons. We refer to Appendix B for proofs of the results in this section, including a specialized discussion for the simple case of a cyclic group.

**Assumptions on $x$.** Our analysis requires a few mild assumptions on the encoding vector $x$.

- Mean centered, $\widehat{x}[\rho_{\text{triv}}] = \langle x, \mathbf{1} \rangle = 0$.
- For all $\rho \in \mathcal{I}(G)$, $\widehat{x}[\rho]$ is either invertible or zero.
- For $\rho \in \mathcal{I}(G)$ such that $\widehat{x}[\rho] \neq 0$, the quantities on the right-hand side of (13) are distinct.

Intuitively, the first condition centers the data, which is necessary since the network includes no bias term. The second and third conditions hold for almost all $x \in \mathbb{R}^{|G|}$ and ensure non-degeneracy and separation in the Fourier coefficients of $x$, leading to a clear step-wise learning behavior.

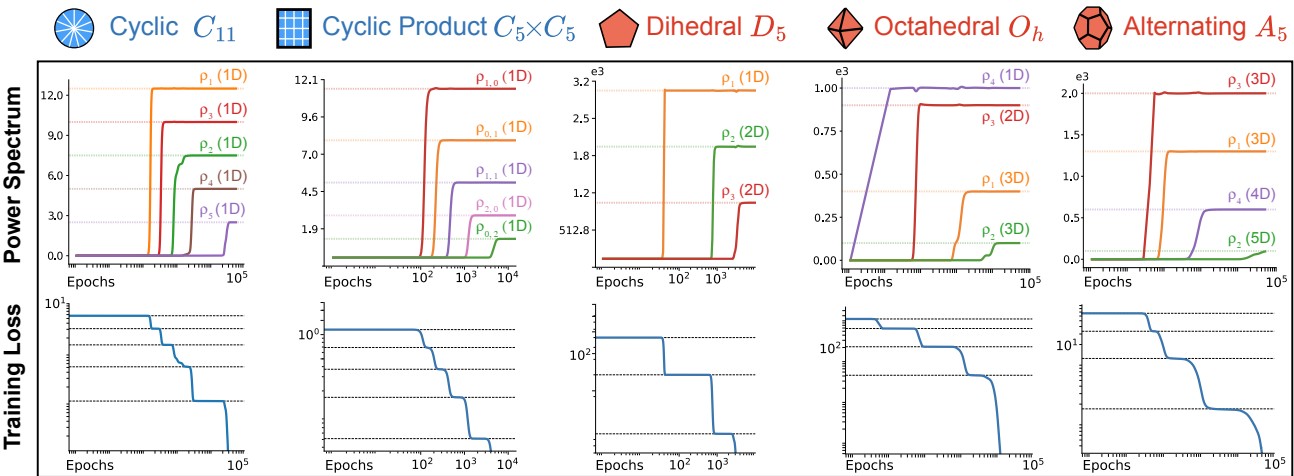

*Figure 3.* **Binary composition on Abelian and Non-Abelian groups.** A two-layer quadratic MLP learns to perform binary group composition task on Abelian and non-Abelian groups by learning the irreducible representations of the group one at a time in order of their importance to the encoding of the group as prescribed in Equation (14). Experimental details are given in Appendix D.1.

**Sigma-pi-sigma decomposition.** Throughout our analysis, we decompose the per-neuron function $f(x_{\mathbf{g}}; \theta_i)$ into two terms:

$$f(x_{\mathbf{g}}; \theta_i)^{(\times)} = w_i\, k! \prod_{j=1}^{k} \langle u_{i,j}, x_{g_j} \rangle, \qquad (10)$$

$$f(x_{\mathbf{g}}; \theta_i)^{(+)} = f(x_{\mathbf{g}}; \theta_i) - f^{(\times)}(x_{\mathbf{g}}, \theta_i). \qquad (11)$$

The term $f(x_{\mathbf{g}}; \theta_i)^{(\times)}$ captures interactions among all the inputs $x_{g_1}, \ldots, x_{g_k}$ and corresponds to a unit in a *sigma-pi-sigma* network (Li, 2003). We will find that this term plays the fundamental role in learning the group composition task. The term $f(x_{\mathbf{g}}; \theta_i)^{(+)}$ will turn out to be extraneous to the task and multiple neurons will need to collaborate to cancel it out. As we demonstrate in Sections 4.3 and 5, different architectures employ distinct mechanisms to cancel this term while retaining the interaction term, producing substantial differences in parameter and computational efficiency. In Appendix C.2, we discuss how smooth non-polynomial activations can access analogous interaction terms through their Taylor expansions near the origin.

**Inductive setup.** We will proceed by induction on the iterations of AGF. To this end, we fix $t \in \mathbb{Z}_{\geq 1}$, and assume that after the $(t-1)^{\text{th}}$ iteration of AGF, the function computed by the active neurons $\mathcal{A}$ is, for $\mathbf{g} \in G^k, h \in G$:

$$f(x_{\mathbf{g}}; \Theta_{\mathcal{A}})[h] = \frac{1}{|G|} \sum_{\rho \in \mathcal{I}^{t-1}} \left\langle \rho(g_1 \cdots g_k h)^{\dagger}, \widehat{x}[\rho] \right\rangle_{\rho}. \qquad (12)$$

Here, $\mathcal{I}^{t-1} \subseteq \mathcal{I}(G)$ is the set of irreps already learned by the network, which we assume is closed under conjugation: if $\rho \in \mathcal{I}^{t-1}$, then $\overline{\rho} \in \mathcal{I}^{t-1}$. If $\mathcal{I}^{t-1} = \mathcal{I}(G)$, then $f(x_{\mathbf{g}}; \Theta_{\mathcal{A}}) = x_{g_1 \cdots g_k}$, indicating the model has perfectly learned the task. At vanishing initialization $\mathcal{I}^0 = \{\rho_{\text{triv}}\}$.

**Utility maximization.** By using the Fourier transform over groups, we prove the following.

**Theorem 4.1.** *At the $t^{\text{th}}$ iteration of AGF, the utility function of $f(\bullet, \theta)$ for a single neuron parametrized by $\theta = (u_1, \ldots, u_k, w)$ coincides with the utility of $f(\bullet, \theta)^{(\times)}$. Moreover, under the constraint $\|\theta\| = 1$, this utility is maximized when the Fourier coefficients of $u_1, \ldots, u_k, w$ are concentrated in $\rho_*$ and $\overline{\rho_*}$, where*

$$\rho_* = \underset{\rho \in \mathcal{I}(G) \setminus \mathcal{I}^{t-1}}{\arg\max} \frac{\|\widehat{x}[\rho]\|_{\text{op}}^{k+1}}{(C_\rho n_\rho)^{\frac{k-1}{2}}}. \qquad (13)$$

*Here, $\|\bullet\|_{\text{op}}$ denotes the operator norm, and $C_\rho = 1$ if $\rho$ is real ($\overline{\rho} = \rho$), and $C_\rho = 2$ otherwise. That is, there exist matrices $s_1, \ldots, s_k, s_w \in \mathbb{C}^{n_{\rho_*} \times n_{\rho_*}}$ such that, for $g \in G$,*

$$u_j[g] = \operatorname{Re} \operatorname{Tr}(\rho_*(g) s_j), \quad w[g] = \operatorname{Re} \operatorname{Tr}(\rho_*(g) s_w). \qquad (14)$$

Put simply, the utility maximizers are real parts of complex linear combinations of the matrix entries of $\rho_*$. Thus, as anticipated, neurons "align" to $\rho_*$ during this phase.

A notable consequence of Theorem 4.1 is a systematic bias toward learning lower-dimensional irreps, an effect that is amplified with sequence length. This bias is particularly transparent for a one-hot encoding, where $\|\widehat{x}[\rho]\|_{\text{op}} = 1$ for all $\rho$, yet the utility still favors smaller $n_\rho$ as $k$ grows. Our theory thus establishes a form of *strong universality* hypothesized in Chughtai et al. (2023)—that representations are acquired from lower- to higher-dimensional irreps—and explains why this ordering was difficult to detect empirically: for $k = 2$ the effect is subtle, but it becomes pronounced as sequence length increases (see Appendix D.2).

**Cost minimization.** To study cost minimization, we assume that after the utility has been maximized at the $t^{\text{th}}$

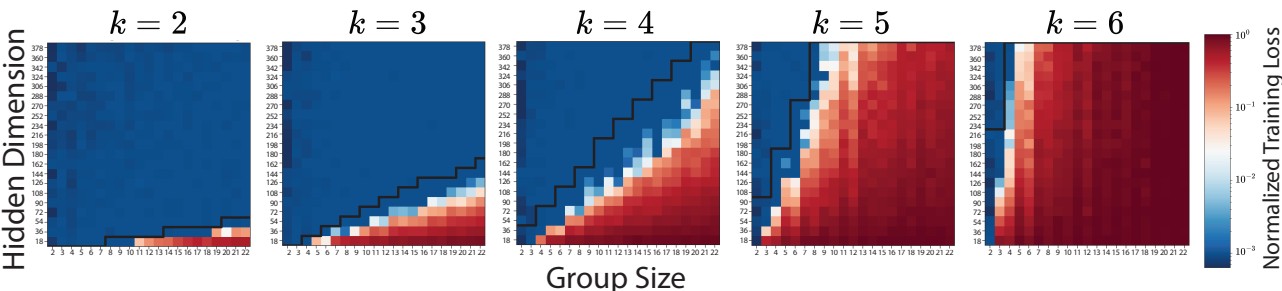

Figure 4. **Two-layer networks need exponential width.** We report results for two-layer networks with quadratic activations trained on the sequential group composition task with the cyclic group over 2205 training runs, spanning 21 group sizes, 21 hidden widths, and 5 sequence lengths. Heatmap colors indicate the normalized training loss, defined as the final loss divided by the initial loss. Training is run until the network achieves a 99.9% loss reduction or reaches $10^6$ optimization steps. The solid black line shows the theoretical lower bound for perfect learning, $H \geq (k+1)2^{k-1}\lfloor \frac{|G|}{2} \rfloor$, derived in Section 4.3. Experimental details can be found in Appendix D.3.

iteration, a group $\mathcal{A}_t$ of $N \leq H$ neurons activates simultaneously. Due to Theorem 4.1, these neurons are aligned to $\rho_*$, i.e., are in the form of (14). Inductively, we assume that the neurons activated in the previous iterations are aligned to irreps in $\mathcal{I}^{t-1}$, and are at an optimal configuration. We then make the following simplifying assumption:

**Assumption 4.2.** *During cost minimization, the newly-activated neurons remain aligned to $\rho_*$.*

This is a natural assumption, that we empirically observe in practice—see Appendix C.1 for a discussion. Thus, we can restrict the cost minimization problem to the space of $\rho_*$-aligned neurons and solve this problem. In particular, we show that, for a large enough number of neurons $N$, a solution must necessarily satisfy $f(x_{\mathbf{g}}; \Theta_{\mathcal{A}_t})^{(+)} = 0$, i.e., the MLP implements a sigma-pi-sigma network.

**Theorem 4.3.** *Under Assumption 4.2, the following bound holds for the loss restricted to the newly-activated neurons:*

$$\mathcal{L}(\Theta_{\mathcal{A}_t}) \geq \frac{1}{2}\left( \|x\|^2 - \frac{C_{\rho_*}}{|G|}\|\widehat{x}[\rho_*]\|^2_{\rho_*} \right). \quad (15)$$

*For $N \geq (k+1)2^k n_{\rho_*}^{k+1}$, the bound is achievable. In this case, it must hold that $f(x_{\mathbf{g}}; \Theta_{\mathcal{A}_t})^{(+)} = 0$, and the function computed by the neurons is, for $\mathbf{g} \in G^k, h \in G$:*

$$f(x_{\mathbf{g}}; \Theta_{\mathcal{A}_t})[h] = \frac{C_{\rho_*}}{|G|}\mathrm{Re}\left\langle \rho_*(g_1 \cdots g_k h)^\dagger, \widehat{x}[\rho_*] \right\rangle_{\rho_*}. \quad (16)$$

Equation (16) concludes the proof by induction. Once the loss has been minimized, the newly-activated neurons $\mathcal{A}_t$, together with the neurons activated in the previous iterations of AGF, will compute a sum in the form of (12), but with the index set given by $\mathcal{I}^t := \mathcal{I}^{t-1} \cup \{\rho_*, \overline{\rho_*}\}$.

### 4.3. Limits of Width: Coordinating Neurons

Theorem 4.3 establishes that an exponential number of neurons is sufficient to exactly learn the sequential group composition task. Our construction of solutions is explicit; in

order to extract sigma-pi-sigma terms from the MLP, we rely on a decomposition of the square-free monomial:

$$z_1 \cdots z_k = \frac{1}{k! \, 2^k} \sum_{\varepsilon \in \{\pm 1\}^k} \left(\prod_{i=1}^k \varepsilon_i\right) \sigma\left(\sum_{i=1}^k \varepsilon_i z_i\right). \quad (17)$$

When $\sigma(z) = z^k$, this is an instance of a *Waring decomposition*, expressing the monomial as a sum of $k^{\text{th}}$ powers. We conclude that $2^k$ neurons can implement a sigma-pi-sigma neuron. We then show that $(k+1)n_\rho^{k+1}$ sigma-pi-sigma neurons can achieve the bound in Theorem 4.3. This leads to a *sufficient* width condition to represent the task exactly:

$$H \geq (k+1)2^k \sum_{\rho \in \mathcal{J}(G)} n_\rho^{k+1}, \quad (18)$$

where $\mathcal{J}(G)$ denotes the non-trivial irreps of $G$, modulo conjugation. For the cyclic group with monomial activation $\sigma(z) = z^k$, this reduces to $H \geq (k+1)2^{k-1}\left\lfloor \frac{|G|}{2} \right\rfloor$, consistent with the empirical scaling in Figure 4. In fact, the exponential factor is optimal in general—see Remark B.7.

This explicit construction both quantifies the width required for perfect performance and clarifies the limitations of narrow networks, which cannot coordinate enough neurons to cancel all extraneous terms. Empirically, we observe an intermediate regime in which the network lacks sufficient capacity for exact learning yet attains strong performance by finding partial solutions. These regimes are often associated with unstable dynamics, potentially related to recent results of Martinelli et al. (2025), who show how pairs of neurons can collaborate to approximate gated linear units at the "edge of stability."

## 5. Benefits of Depth: Leveraging Associativity

As established in Section 4.3 and illustrated in Figure 4, while two-layer MLPs can perfectly learn the group composition task, they scale poorly in both parameter and sample

complexity—requiring exponentially many hidden neurons with respect to sequence length $k$. This raises a natural question: *can deeper architectures, built for sequential computation, discover more efficient compositional solutions?*

We partially answer this question by showing that recurrent and multilayer architectures can, in principle, exploit the associativity of group operations to compose intermediate representations, leading to dramatically more efficient solutions. As in the two-layer setting, our constructions are organized by the decomposition of the task into irreducible representations. However, unlike in Section 4, our results here are constructive rather than dynamical: we show that efficient solutions exist, but we do not analyze whether or how gradient descent converges to them. We do find partial empirical evidence that trained recurrent networks recover structure consistent with our construction; see Figure 5.

### 5.1. RNNs Learn to Compose Sequentially

We first consider a recurrent neural network (RNN) with a quadratic nonlinearity $\sigma(z) = z^2$, that computes:

$$h^{(2)} = \sigma(W_{\text{in}} x_{g_1} + W_{\text{drive}} x_{g_2}),$$
$$h^{(i)} = \sigma(W_{\text{mix}} h^{(i-1)} + W_{\text{drive}} x_{g_i}), \qquad (19)$$
$$f_{\text{rnn}}(x_{\mathbf{g}}; \Theta) = W_{\text{out}} h^{(k)}.$$

Here $W_{\text{in}}, W_{\text{drive}} \in \mathbb{R}^{H \times |G|}$ embed the inputs $x_{g_i}$ into a hidden representation, $W_{\text{mix}} \in \mathbb{R}^{H \times H}$ mixes the hidden representation between steps, and $W_{\text{out}} \in \mathbb{R}^{|G| \times H}$ unembeds the final hidden representation into a prediction. This RNN is an instance of an *Elman network* (Elman, 1990) and when $k = 2$, it reduces to a two-layer MLP with a quadratic non-linearity, as discussed in Section 4.

Now, we show that $f_{\text{rnn}}$ can learn the group composition task without requiring a hidden width that grows exponentially with $k$, by explicitly constructing a solution within this architecture. The RNN will exploit associativity to compute the group composition sequentially:

$$g_1 \cdots g_k = (((((g_1 \cdot g_2) \cdot g_3) \cdot g_4) \cdots g_{k-1}) \cdot g_k). \quad (20)$$

We will achieve this by combining two-layer MLPs. To this end, let $W_{\text{in}}^{\text{mlp}}, W_{\text{out}}^{\text{mlp}}$ be weights for an MLP with activation $\sigma(z) = z^2$ that perfectly learns the binary group composition task, as constructed in Section 4. Split $W_{\text{in}}^{\text{mlp}} = [W_{\text{left}}^{\text{mlp}} \mid W_{\text{right}}^{\text{mlp}}]$ columns-wise into the sub-matrices corresponding to the two group inputs, and set:

$$W_{\text{in}} = W_{\text{left}}^{\text{mlp}}, \qquad W_{\text{drive}} = W_{\text{right}}^{\text{mlp}},$$
$$W_{\text{mix}} = W_{\text{left}}^{\text{mlp}} W_{\text{out}}^{\text{mlp}}, \qquad W_{\text{out}} = W_{\text{out}}^{\text{mlp}}. \qquad (21)$$

By construction, the RNN with these weights solves the task sequentially, in the spirit of Equation (20); for each $i$,

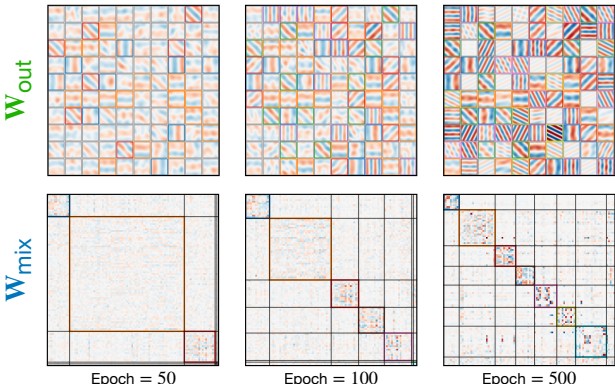

Epoch = 50     Epoch = 100     Epoch = 500

*Figure 5.* **Empirical evidence for sequential composition in recurrent networks.** For $G = C_n \times C_n$, trained RNNs acquire irreducible representations one at a time: $W_{\text{out}}$ specializes to irrep-aligned modes, while $W_{\text{mix}}$ becomes approximately block diagonal in the same decomposition. This provides partial empirical evidence that gradient descent can find recurrent solutions consistent with our irrep-based construction.

we have $W_{\text{out}} h^{(i)} = x_{g_1 \cdots g_i}$. As a result, the RNN is able to learn the task with $H = \mathcal{O}(\sum_{\rho \in \mathcal{I}(G)} n_\rho^3) = \mathcal{O}(|G|^{\frac{3}{2}})$ hidden neurons, which is constant in the sequence length $k$.

An interesting property of our construction is that $W_{\text{mix}}$ is permutation-similar to a *block-diagonal* matrix, with each block corresponding to a given irrep of $G$. This follows from Schur's orthogonality relations (see Appendix A), since the columns of $W_{\text{out}}^{\text{mlp}}$ and the rows of $W_{\text{left}}^{\text{mlp}}$ are aligned with irreps. In other words, $W_{\text{mix}}$ learns to only mix hidden representations corresponding to the same irrep.

### 5.2. Deep MLPs Learn to Compose in Parallel

We now consider a multilayer feedforward architecture. As in the RNN, depth allows the group composition task to be implemented using only binary interactions, eliminating the need for exponential width. Here, these interactions are arranged in parallel along a balanced tree. For simplicity, we assume $k = 2^L$ and consider a depth-$L$ multilayer MLP of the form

$$h^{(\ell)} = \sigma(W^{(\ell)} h^{(\ell-1)}), \qquad \ell = 1, \ldots, L,$$
$$f_{\text{mlp}}(x_{\mathbf{g}}; \Theta) = W^{(L+1)} h^{(L)}, \qquad (22)$$

where $h^{(0)} = x_{\mathbf{g}}$ and $\sigma(z) = z^2$ is applied elementwise. The hidden widths decrease geometrically: at level $\ell$, the representation consists of $k/2^\ell$ intermediate group elements, each embedded in a $H$-dimensional hidden space. As in Section 5.1, when $k = 2$ this architecture reduces to the two-layer MLP studied in Section 4.

We now show that $f_{\text{mlp}}$ can learn the group composition task with $H = \mathcal{O}(|G|^{\frac{3}{2}})$ by explicitly constructing a solution within this architecture. Like the RNN, our construction

will perform $k - 1$ binary group compositions; however, it does so in parallel along a balanced tree, reducing the depth of the computation from $k$ steps in time to $\log k$ layers:

$$g_1 \cdots g_k = \big((g_1 \cdot g_2) \cdot (g_3 \cdot g_4)\big) \cdots (g_{k-1} \cdot g_k). \quad (23)$$

As in Section 5.1, we use the building blocks $W_{\mathrm{in}}^{\mathrm{mlp}} \in \mathbb{R}^{H \times 2|G|}$ and $W_{\mathrm{out}}^{\mathrm{mlp}} \in \mathbb{R}^{|G| \times H}$ of a two-layer MLP that perfectly learns binary group composition and construct,

$$W_{\mathrm{merge}} = W_{\mathrm{in}}^{\mathrm{mlp}} \big(\mathbf{I}_2 \otimes W_{\mathrm{out}}^{\mathrm{mlp}}\big) \in \mathbb{R}^{H \times 2H}. \quad (24)$$

We then set the weights of the depth-$L$ multilayer MLP with $k = 2^L$ to be block-diagonal lifts of these maps:

$$\begin{aligned} W^{(1)} &= \mathbf{I}_{k/2} \otimes W_{\mathrm{in}}^{\mathrm{mlp}}, \\ W^{(\ell)} &= \mathbf{I}_{k/2^\ell} \otimes W_{\mathrm{merge}}, \qquad \ell = 2, \dots, L, \quad (25) \\ W^{(L+1)} &= W_{\mathrm{out}}^{\mathrm{mlp}}. \end{aligned}$$

As in Section 5.1, because $W_{\mathrm{in}}^{\mathrm{mlp}}$ and $W_{\mathrm{out}}^{\mathrm{mlp}}$ are aligned with the irreducible representations of $G$, the effective merge operator $W_{\mathrm{merge}}$ is permutation-similar to a block-diagonal matrix with blocks indexed by irreps. As a result, each irrep is composed independently throughout the tree.

### 5.3. Transformers Can Learn Algebraic Shortcuts

Given the prominence of the transformer architecture, it is natural to ask how such models solve the sequential group composition task. Related work by Liu et al. (2022) studies how transformers simulate finite-state semiautomata, a generalization of group composition. They show that logarithmic-depth transformers can simulate all semiautomata, and that for the class of *solvable* semiautomata, constant-depth simulators exist at the cost of increased width. Their logarithmic-depth construction is essentially the parallel divide-and-conquer strategy underlying our multilayer MLP construction. Their constant-depth construction instead relies on decompositions of the underlying algebraic structure, suggesting that analogous constant-depth shortcuts should exist for sequential group composition over *solvable groups*. Characterizing these algebraic shortcuts explicitly, and understanding when gradient-based training biases transformers toward such shortcuts rather than the sequential or parallel composition strategies, is an interesting question. A recent work in this direction (McCracken et al., 2026) discussed how the Chinese Remainder Theorem–which can be seen as an instance of a decomposition/filtration—can be exploited for shortcuts in the modular addition task.

## 6. Discussion

This work was motivated by a central question in modern deep learning: how do neural networks trained over sequences acquire the ability to perform structured operations, such as arithmetic, geometric, and algorithmic computation? To gain insight into this question, we introduced the *sequential group composition* task and showed that this task can be order-sensitive, provably requires nonlinear architectures (Section 3), admits tractable feature learning (Section 4), and reveals an interpretable benefit of depth (Section 5).

**Limitations.** Our theory in Section 4 is developed in a controlled setting: finite groups, orbit-based encodings, polynomial activations, mean-squared loss, full-batch training, and vanishing initialization. These assumptions make the feature-learning dynamics analytically tractable, but they also limit the direct scope of our results. Extending the analysis beyond this setting remains an important direction for future work; see Appendix C for preliminary theoretical and empirical evidence beyond some of these assumptions. A second limitation is that our results for deep networks in Section 5 are constructive rather than dynamical: they show that these architectures can exploit associativity to obtain efficient solutions, but do not prove that gradient descent finds them. While the RNN experiment in Figure 5 provides partial empirical evidence that gradient descent can recover structure consistent with our construction, substantially more evidence would be needed to establish that such associativity-based solutions are found reliably. Developing an AGF-style dynamical theory for recurrent and deep networks remains a major open challenge (Kunin et al., 2025).

**From groups to semiautomata.** Groups are only one corner of algebraic computation: they correspond to reversible dynamics, where each input symbol induces a bijection on the state space. More generally, a *semiautomaton* is a triple $(Q, \Sigma, \delta)$, where $Q$ is a set of states, $\Sigma$ is an alphabet, and $\delta \colon Q \times \Sigma \to Q$ is a transition map. The collection of all maps $\delta(\cdot, \sigma)$ forms a *transformation semigroup* on $Q$. Unlike groups, this semigroup can contain both reversible permutation operations and irreversible operations such as resets. Extending our framework from groups to semiautomata would therefore allow us to study how networks learn both reversible and irreversible computations.

**From semiautomata to formal grammars.** Semiautomata generate exactly the class of regular languages, but many symbolic tasks require richer structures. A *formal grammar* $(V, \Sigma, R, S)$ is defined with nonterminals $V$, terminals $\Sigma$, production rules $R$, and start symbol $S$. Restricting the form of the rules recovers the Chomsky hierarchy: regular grammars (equivalent to finite automata), context-free grammars (captured by pushdown automata).This marks a shift from *associativity* as the key inductive bias to *recursion*: networks must learn to encode and apply hierarchical rules.

Taken together, these extensions raise the question of how far our dynamical analysis of sequential group composition can be extended toward language-related tasks.

## Acknowledgements

We thank Jason D. Lee, Flavio Martinelli, and Eric J. Michaud for helpful conversations. This work was partially supported by the Wallenberg AI, Autonomous Systems and Software Program (WASP) funded by the Knut and Alice Wallenberg Foundation, and by the Miller Institute for Basic Research in Science, University of California, Berkeley. Nina is partially supported by NSF grant 2313150 and the NSF CAREER Award 240158. Francisco is supported by NSF grant 2313150. Adele is supported by NSF GRFP and NSF grant 240158.

## Impact Statement

This paper presents work whose goal is to advance the field of Machine Learning. There are many potential societal consequences of our work, none which we feel must be specifically highlighted here.

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

# A. Additional Background on Harmonic Analysis over Groups

Here, we summarize the main properties of the Fourier transform over (finite) groups (see Definition 3.3):

- *Diagonalization.* The matrix $F$ simultaneously block-diagonalizes $\lambda(g)$ for all $g \in G$:

$$F^\dagger \lambda(g) F = \bigoplus_{\rho \in \mathcal{I}(G)} \frac{|G|}{n_\rho} \underbrace{\rho(g) \oplus \cdots \oplus \rho(g)}_{n_\rho \text{ copies}}. \tag{26}$$

  Constants $|G|$ and $n_\rho$ in Equation (26) are sometimes absorbed into the definition of $F$; here they are included in the Hermitian product for convenience.

- *Convolution theorem.* For $x, y \in \mathbb{C}^G$, the *group convolution* $\star : \mathbb{C}^G \times \mathbb{C}^G \to \mathbb{C}^G$ is defined by

$$(x \star y)[g] = x^\dagger \lambda(g)^\mathsf{T} y = \sum_{h \in G} \overline{x[h]}\, y[gh]. \tag{27}$$

  That is, $(x \star y)[g]$ computes the inner product between $x$ and the left-translated version of $y$ under the regular representation $\lambda(g)$. Then, for every $\rho \in \mathcal{I}(G)$,

$$\widehat{x \star y}[\rho] = \widehat{x}[\rho]^\dagger \widehat{y}[\rho]. \tag{28}$$

  In other words, convolution in the group domain corresponds to matrix multiplication in the Fourier domain.

- *Plancherel theorem.* For $\rho \in \mathcal{I}(G)$ and $A, B \in \mathbb{C}^{n_\rho \times n_\rho}$, define the normalized Frobenius Hermitian product $\langle A, B \rangle_\rho = n_\rho \operatorname{Tr}(A^\dagger B)$, which induces the inner product $\frac{1}{|G|} \sum_{\rho \in \mathcal{I}(G)} \langle \cdot, \cdot \rangle_\rho$ over $\bigoplus_{\rho \in \mathcal{I}(G)} \mathbb{C}^{n_\rho \times n_\rho}$. With respect to this inner product and the standard Hermitian inner product on $\mathbb{C}^G$, the Fourier transform is an invertible unitary operator between $\mathbb{C}^G$ and its frequency-domain. In other words, for all $x, y \in \mathbb{C}^G$,

$$\langle x, y \rangle = \frac{1}{|G|} \sum_{\rho \in \mathcal{I}(G)} \langle \widehat{x}[\rho], \widehat{y}[\rho] \rangle_\rho. \tag{29}$$

- *Schur orthogonality relations.* Explicitly, for two irreducible representations $\rho_1, \rho_2 \in \mathcal{I}(G)$ and two matrices $A_1 \in \mathbb{C}^{n_{\rho_1} \times n_{\rho_1}}$, $A_2 \in \mathbb{C}^{n_{\rho_2} \times n_{\rho_2}}$, it holds that:

$$\sum_{g \in G} \left\langle \rho_1(g)^\dagger, A_1 \right\rangle_{\rho_1} \left\langle \rho_2(g)^\dagger, A_2 \right\rangle_{\rho_2} = \begin{cases} |G| \left\langle \overline{A_1}, A_2 \right\rangle_{\rho_1} & \rho_1 = \overline{\rho_2}, \\ 0 & \rho_1 \neq \overline{\rho_2}. \end{cases} \tag{30}$$

- *Properties of the character.* The *character* of a representation $\rho$ is the class function $\chi_\rho(g) := \operatorname{Tr}(\rho(g))$. A useful fact is that the group Fourier transform of $\chi_\rho$ satisfies

$$\widehat{\chi_\rho}[\rho'] = \begin{cases} \frac{|G|}{n_\rho} I, & \rho = \rho', \\ 0, & \rho \neq \rho'. \end{cases} \tag{31}$$

## A.1. Non-linearity of the Task

We now prove that the sequential group composition task can not be implemented by a linear map.

**Lemma A.1.** *Assume that $\widehat{x}[\rho_{\mathrm{triv}}] = \langle x, \mathbf{1} \rangle = 0$, but $x \neq 0$. There is no linear map $\mathbb{R}^{k|G|} \to \mathbb{R}^{|G|}$ sending $x_{\mathbf{g}}$ to $x_{g_1 \cdots g_k}$ for all $\mathbf{g} \in G^k$.*

*Proof.* Suppose that $L \colon \mathbb{R}^{k|G|} \to \mathbb{R}^{|G|}$ is a linear map (i.e., a matrix) sending $x_{\mathbf{g}}$ to $x_{g_1 \cdots g_k}$ for all $\mathbf{g} \in G^k$. By linearity, we can split this map as $L x_{\mathbf{g}} = \sum_{i=1}^k L_i x_{g_i}$ for opportune $|G| \times |G|$ matrices $L_i$. Since $x \neq 0$, for all $\mathbf{g} \in G^k$, we have that $0 \neq \|x_{g_1 \cdots g_k}\|^2 = \langle x_{g_1 \cdots g_k}, L x_{\mathbf{g}} \rangle = \sum_{i=1}^k \langle x_{g_1 \cdots g_k}, L_i x_{g_i} \rangle$. But since $\langle x, \mathbf{1} \rangle = 0$, we have

$$\sum_{\mathbf{g} \in G^k} \sum_{i=1}^k \langle x_{g_1 \cdots g_k}, L x_{g_i} \rangle = \sum_{i=1}^k \sum_{g_i \in G} \left\langle \sum_{\mathbf{g}' \in G^{k-1}} x_{g_1 \cdots g_k}, L_i x_{g_i} \right\rangle = 0, \tag{32}$$

where $\mathbf{g}'$ contains all the indices different from $i$. This leads to a contradiction. $\qquad\square$

# B. Proofs of Feature Learning in Two-layer Networks (Section 4)

## B.1. Utility Maximization

As explained in Section 4.2 we assume, inductively, that after the $t-1$ iterations of AGF, the function computed by the active neurons is, for $\mathbf{g} \in G^k, h \in G$:

$$f(x_{\mathbf{g}}; \Theta_{\mathcal{A}})[h] = \frac{1}{|G|} \sum_{\rho \in \mathcal{I}^{t-1}} \left\langle \rho(g_1 \cdots g_k h)^{\dagger}, \widehat{x}[\rho] \right\rangle_{\rho}, \tag{33}$$

where $\mathcal{I}^{t-1} \subseteq \mathcal{I}(G)$ is closed under conjugation.

We begin by proving a useful identity.

**Lemma B.1.** *For* $\mathbf{g} \in G^k$, *we have:*

$$\sum_{\mathbf{g} \in G^k} \langle w, x_{g_1 \cdots g_k} \rangle \prod_{i=1}^{k} \langle u_i, x_{g_i} \rangle = \frac{1}{|G|} \sum_{\rho \in \mathcal{I}(G)} \langle \widehat{u_1}[\rho]^{\dagger} \widehat{x}[\rho] \cdots \widehat{u_k}[\rho]^{\dagger} \widehat{x}[\rho], \widehat{w}[\rho]^{\dagger} \widehat{x}[\rho] \rangle_{\rho}. \tag{34}$$

*Proof.* Note that $\langle u_i, x_{g_i} \rangle = (u_i \star x)[g_i]$. We can rewrite the left-hand side of (34) as:

$$\sum_{\mathbf{g} \in G^k} \langle w, x_{g_1 \cdots g_k} \rangle \prod_{i=1}^{k} \langle u_i, x_{g_i} \rangle = \sum_{\mathbf{g}' \in G^{k-1}} \prod_{i=1}^{k-1} (u_i \star x)[g_i] \sum_{g_k \in G} (w \star x)[g_1 \cdots g_k] (u_k \star x)[g_k]$$

$$= \sum_{\mathbf{g}' \in G^{k-1}} \left( (u_k \star x) \star (w \star x) \right)[g_1 \cdots g_{k-1}] \prod_{i=1}^{k-1} (u_i \star x)[g_i], \tag{35}$$

where $\mathbf{g}' = (g_1, \ldots, g_{k-1})$. By iterating this argument, we conclude that the above expression equals

$$\langle u_1 \star x, \; (\cdots (u_{k-1} \star x) \star ((u_k \star x) \star (w \star x))) \rangle. \tag{36}$$

By Plancharel (29), this scalar product can be phrased as a sum of scalar products between the Fourier coefficients. The desired expression (34) follows then from the convolution theorem (28) applied, iteratively, to the convolutions appearing in (36). $\qquad \square$

We now compute the utility function at the next iteration of AGF.

**Lemma B.2.** *At the* $t^{\text{th}}$ *iteration of AGF, the utility function of* $f(\bullet, \theta)$ *for a single neuron parametrized by* $\theta = (u_1, \ldots, u_k, w)$ *coincides with the utility of* $f(\bullet, \theta)^{(\times)}$, *and can be expressed as:*

$$\frac{k!}{|G|^{k+1}} \sum_{\mathcal{I}(G) \setminus \mathcal{I}^{t-1}} \langle \widehat{u_1}[\rho]^{\dagger} \widehat{x}[\rho] \cdots \widehat{u_k}[\rho]^{\dagger} \widehat{x}[\rho], \; \widehat{w}[\rho]^{\dagger} \widehat{x}[\rho] \rangle_{\rho}. \tag{37}$$

*Proof.* By the definition of utility and the inductive hypothesis, we have:

$$\mathcal{U}^t(\theta) = \frac{1}{|G|^k} \sum_{\mathbf{g} \in G^k} \sigma \left( \sum_{i=1}^{k} \langle u_i, x_{g_i} \rangle \right) \left( \langle w, x_{g_1 \cdots g_k} \rangle - \frac{1}{|G|} \sum_{\rho \in \mathcal{I}^{t-1}} \langle w, \chi^{\rho}_{g_1 \cdots g_k} \rangle \right), \tag{38}$$

where $\chi^{\rho}[p] = \langle \rho(p)^{\dagger}, \widehat{x}[\rho] \rangle_{\rho}$. We now expand $\sigma(\sum_{i=1}^{k} \langle u_i, x_{g_i} \rangle)$ into a sum of monomials (of degree $\leq k$) in the terms $\langle u_1, x_{g_1} \rangle, \ldots, \langle u_k, x_{g_k} \rangle$. The only monomial where all the group elements $g_1, \ldots, g_k$ appear is $k! \prod_{i=1}^{k} \langle u_i, x_{g_i} \rangle$. For any other monomial, the term $\langle w, \chi^{\rho}_{g_1 \cdots g_k} \rangle$ will vanish, since $\sum_{g \in G} \rho(g) = 0$. Thus, (38) reduces to the utility of $f(\bullet, \theta)^{(\times)}$, i.e.:

$$\frac{k!}{|G|^k} \sum_{\mathbf{g} \in G^k} \prod_{i=1}^{k} \langle u_i, x_{g_i} \rangle \left( \langle w, x_{g_1 \cdots g_k} \rangle - \frac{1}{|G|} \sum_{\rho \in \mathcal{I}^{t-1}} \langle w, \chi^{\rho}_{g_1 \cdots g_k} \rangle \right). \tag{39}$$

We can expand the above expression by using Lemma B.1. For each $\rho \in \mathcal{I}^{t-1}$, the term containing $\langle w, \chi^{\rho}_{g_1 \cdots g_k} \rangle$ will cancel out the summand indexed by $\rho$ in the right-hand side of (34). In conclusion, (39) reduces to the desired expression (37). $\qquad \square$

**Theorem B.3.** *Let*

$$\rho_* = \operatorname*{argmax}_{\rho \in \mathcal{I}(G) \setminus \mathcal{I}^{t-1}} (n_\rho C_\rho)^{\frac{1-k}{2}} \|\widehat{x}[\rho]\|_{\mathrm{op}}^{k+1}, \tag{40}$$

*where $\| \bullet \|_{\mathrm{op}}$ denotes the operator norm, and $C_\rho$ is a coefficient which equals to $1$ if $\rho$ is real, and to $2$ otherwise. The unit parameter vectors $\theta = (u_1, \ldots, u_k, w)$ that maximize the utility function $\mathcal{U}^t$ take the form, for $g \in G$,*

$$
\begin{aligned}
u_j[g] &= \operatorname{Re} \left\langle \rho_*(g)^\dagger, s_j \right\rangle_{\rho_*}, \\
w[g] &= \operatorname{Re} \left\langle \rho_*(g)^\dagger, s_w \right\rangle_{\rho_*},
\end{aligned}
\tag{41}
$$

*where $s_j, s_w \in \mathbb{C}^{n_{\rho_*} \times n_{\rho_*}}$ are matrices. When $\rho$ is real ($\rho_* = \overline{\rho_*}$), these matrices are real.*

Note that the above argmax is well-defined since, by our assumptions on $x$ (see Section 4.2), the maximizer of $\|\widehat{x}[\rho]\|_\rho$ is unique up to conjugate.

*Proof.* For simplicity, denote $u_0 = w$. Using Lemma B.2 and by Plancharel, the optimization problem can be rephrased in terms of the Fourier transform as:

$$
\begin{aligned}
\text{maximize} \quad & \frac{k!}{|G|^{k+1}} \sum_{\rho \in \mathcal{I}(G) \setminus \mathcal{I}^{t-1}} n_\rho \operatorname{Tr} \left( \widehat{x}[\rho]^\dagger \widehat{u_1}[\rho] \cdots \widehat{x}[\rho]^\dagger \widehat{u_k}[\rho] \widehat{u_0}[\rho]^\dagger \widehat{x}[\rho] \right) \\
\text{subject to} \quad & \sum_{i=0}^{k} \|u_i\|^2 = \frac{1}{|G|} \sum_{\rho \in \mathcal{I}(G)} \sum_{i=0}^{k} \|\widehat{u_i}[\rho]\|_\rho^2 = 1.
\end{aligned}
\tag{42}
$$

Recall that $\mathcal{I}^{t-1}$ is assumed to be closed by conjugation. Let $\mathcal{J} \subseteq \mathcal{I}(G)$ be a set of representatives for irreps up to conjugate. Up to the multiplicative constant, the utility becomes:

$$\sum_{\rho \in \mathcal{J} \setminus \mathcal{I}^{t-1}} n_\rho C_\rho \operatorname{Re} \operatorname{Tr} \left( \widehat{x}[\rho]^\dagger \widehat{u_1}[\rho] \cdots \widehat{x}[\rho]^\dagger \widehat{u_k}[\rho] \widehat{u_0}[\rho]^\dagger \widehat{x}[\rho] \right). \tag{43}$$

Given an irrep $\rho$, define the coefficient $\alpha_\rho$ as $\alpha_\rho^2 = \frac{C_\rho}{|G|} \sum_{i=0}^{k} \|\widehat{u_i}[\rho]\|_\rho^2$. The constraint becomes $\sum_{\rho \in \mathcal{J}} \alpha_\rho^2 = 1$. Moreover, denote $U_{i,\rho} = \widehat{u_i}[\rho]/\alpha_\rho$, so that

$$\sum_{i=0}^{k} \|U_{i,\rho}\|_\rho^2 = \frac{|G|}{C_\rho}. \tag{44}$$

Let $M_\rho$ be the maximizer of $n_\rho C_\rho \left| \operatorname{Re} \operatorname{Tr} \left( \widehat{x}[\rho]^\dagger U_{1,\rho} \cdots \widehat{x}[\rho]^\dagger U_{k,\rho} U_{0,\rho}^\dagger \widehat{x}[\rho] \right) \right|$ subject to the constraint (44). The original matrix optimization problem is bounded by the scalar optimization problem:

$$
\begin{aligned}
\text{maximize} \quad & \frac{k!}{|G|^{k+1}} \sum_{\rho \in \mathcal{J} \setminus \mathcal{I}^{t-1}} M_\rho \alpha_\rho^{k+1} \\
\text{subject to} \quad & \sum_{\rho \in \mathcal{J}} \alpha_\rho^2 = 1.
\end{aligned}
\tag{45}
$$

This problem is solved, clearly, when $\alpha_\rho$ is concentrated in the irrep $\rho_* \in \mathcal{J} \setminus \mathcal{I}^{t-1}$ maximizing $M_\rho$, meaning that $\alpha_\rho = 0$ for $\rho \neq \rho_*$.

We now wish to describe $M_\rho$. Recall that for complex square matrices $A, B$ we have $|\operatorname{Re} \operatorname{Tr}(AB)| \leq |\operatorname{Tr}(AB)| \leq \|A\|_F \|B\|_F$ and $\|AB\|_F \leq \|A\|_{\mathrm{op}} \|B\|_F \leq \|A\|_F \|B\|_F$, where $\| \bullet \|_F$ denotes the Frobenius norm. By iteratively applying these inequalities, we deduce:

$$n_\rho C_\rho \left| \operatorname{Re} \operatorname{Tr} \left( \widehat{x}[\rho]^\dagger U_{1,\rho} \cdots \widehat{x}[\rho]^\dagger U_{k,\rho} U_{0,\rho}^\dagger \widehat{x}[\rho] \right) \right| \leq n_\rho C_\rho \|\widehat{x}[\rho]\|_{\mathrm{op}}^{k+1} \prod_{i=0}^{k} \|U_{i,\rho}\|_F. \tag{46}$$

Under the constraint (44), the right-hand side of the above expression is maximized when all the $U_{i,\rho}$ have the same Frobenius norm $\|U_{i,\rho}\|_F = (|G|/(C_\rho n_\rho(k+1)))^{\frac{1}{2}}$. This implies that

$$M_\rho \leq n_\rho C_\rho \|\widehat{x}[\rho]\|_{\mathrm{op}}^{k+1} \left( \frac{|G|}{n_\rho C_\rho(k+1)} \right)^{\frac{k+1}{2}} = (n_\rho C_\rho)^{\frac{1-k}{2}} \|\widehat{x}[\rho]\|_{\mathrm{op}}^{k+1} \tag{47}$$

We now show that this bound is realizable. Let $\lambda$ be the largest singular value of $\widehat{x}[\rho]^\dagger$, which coincides with its operator norm, and $p, q$ be the corresponding left and right singular vectors. Define

$$U_{i,\rho} = \left( \frac{|G|}{n_\rho C_\rho(k+1)} \right)^{\frac{1}{2}} qp^\dagger. \tag{48}$$

This is a scaled orthogonal projector. Since $\|qp^\dagger\|_F = 1$, the constraint (44) is satisfied. Moreover, we see that $\widehat{x}[\rho]^\dagger U_{i,\rho} = \lambda(|G|/(n_\rho C_\rho(k+1)))^{\frac{1}{2}} pp^\dagger$. By iteratively applying idempotency of projectors, we see that the left-hand side of (46) equals $n_\rho C_\rho \lambda^{k+1}(|G|/(n_\rho C_\rho(k+1)))^{\frac{k+1}{2}}$, which matches the right-hand side. In conclusion, the bound from (44) is actually an equality. Since the coefficient $(|G|/(k+1))^{\frac{k+1}{2}}$ is constant in $\rho$, the irrep maximizing $M_\rho$ coincides with $\rho_*$, as defined by (40).

Putting everything together, we have constructed maximizers of the original optimization problem (42), and have shown that for all maximizers, the Fourier transform of $u_1, \ldots, u_k, w$ is concentrated in $\rho_*$ and $\overline{\rho_*}$ (which can coincide). The expressions (41) follow by taking the inverse Fourier transform, where $s_i$ and $s_w$ coincide, up to opportune multiplicative constants, with $\widehat{u_i}[\rho_*]$ and $\widehat{w}[\rho_*]$, respectively.

$\square$

## B.2. Cost Minimization

Consider $N$ neurons parametrized by $\Theta_{\mathcal{A}_t} = (\theta_1, \ldots, \theta_N)$, $\theta_i = (u_1^i, \ldots, u_k^i, w^i)$, in the form of (41), i.e.:

$$\begin{aligned}
u_j^i[g] &= \mathrm{Re} \left\langle \rho_*(g)^\dagger, s_j^i \right\rangle_{\rho_*}, \\
w^i[g] &= \mathrm{Re} \left\langle \rho_*(g)^\dagger, s_w^i \right\rangle_{\rho_*},
\end{aligned} \tag{49}$$

where $s_j^i, s_w^i \in \mathbb{C}^{n_{\rho_*} \times n_{\rho_*}}$ are matrices. When $\rho_*$ is real, these matrices are constrained to be real as well. For convenience, we denote $S_j^i = (s_j^i)^\dagger \widehat{x}[\rho_*]$.

As explained in Section 4.2 (Assumption 4.2) we make the assumption that, during cost minimization, the newly-activated neurons stay aligned to $\rho_*$ during cost minimization, i.e., they remain in the form of (49). We expand on this assumption, and provide evidence for it, in Appendix C.1. Now, we can inductively assume that the neurons $\mathcal{A}$ that activated in the previous iterations of AGF are also aligned to the corresponding irreps in $\mathcal{I}^{t-1}$. By looking at the second-layer weights $w_i$, it follows immediately from Schur orthogonality (30) that the loss splits as:

$$\mathcal{L}(\Theta_{\mathcal{A}} \oplus \Theta_{\mathcal{A}_t}) = \mathcal{L}(\Theta_{\mathcal{A}}) + \mathcal{L}(\Theta_{\mathcal{A}_t}). \tag{50}$$

Since the neurons $\mathcal{A}$ have been optimized in the previous iterations of AGF, the gradient of their loss vanishes. Thus, the derivatives of the total loss $\mathcal{L}(\Theta_{\mathcal{A}})$ with respect to parameters of neurons in $\mathcal{A}_t$ coincide with the derivatives of their loss $\mathcal{L}(\Theta_{\mathcal{A}_t})$. Put simply, the newly-activated neurons evolve, under the gradient flow, independently from the previously-activated ones, while the latter remain at equilibrium.

In conclusion, we reduce to solving the cost minimization problem over parameters $\Theta_{\mathcal{A}_t}$ in the form of (49), which we address in the remainder of this section. To this end, we start by showing the following orthogonality property for the sigma-pi-sigma decomposition.

**Lemma B.4.** *The following orthogonality relation holds:*

$$\sum_{\mathbf{g} \in G^k} \left\langle f(x_{\mathbf{g}}; \Theta_{\mathcal{A}_t})^{(\times)}, f(x_{\mathbf{g}}; \Theta_{\mathcal{A}_t})^{(+)} \right\rangle = 0. \tag{51}$$

*Proof.* For $g \in G$, since $\widehat{x_g}[\rho_*] = \widehat{x}[\rho_*]\rho_*(g)$, from Plancharel it follows that:

$$\langle u_j^i, x_g \rangle = \left\langle \widehat{u_j^i}, \widehat{x_g} \right\rangle = \mathrm{Re}\left\langle \rho_*(g)^\dagger, S_j^i \right\rangle_{\rho_*}. \tag{52}$$

By expanding $f(x_\mathbf{g}; \Theta_{\mathcal{A}_t})^{(+)}$ similarly to the proof of Lemma B.2, the product between any of its monomial and the monomials $k! \prod_{h=1}^k \mathrm{Re}\langle \rho_*(g_h)^\dagger, S_h^i \rangle_{\rho_*}$ from $f(x_\mathbf{g}; \Theta_{\mathcal{A}_t})^{(\times)}$ vanishes, since the former will not contain some group element among $g_1, \ldots, g_k$. $\qquad\square$

It follows immediately that loss splits as:

$$\mathcal{L}(\Theta_{\mathcal{A}_t}) = \frac{1}{2|G|^k} \sum_{\mathbf{g} \in G^k} \left\| x_{g_1 \cdots g_k} - f(x_\mathbf{g}; \Theta_{\mathcal{A}_t})^{(+)} - f(x_\mathbf{g}; \Theta_{\mathcal{A}_t})^{(\times)} \right\|^2$$

$$= \frac{1}{2|G|^k} \sum_{\mathbf{g} \in G^k} \left( \left\| f(x_\mathbf{g}; \Theta_{\mathcal{A}_t})^{(+)} \right\|^2 + \left\| f(x_\mathbf{g}; \Theta_{\mathcal{A}_t})^{(\times)} \right\|^2 \right) - \mathcal{U}^1(\Theta_{\mathcal{A}_t}) + \frac{\|x\|^2}{2} \tag{53}$$

$$= \frac{1}{2|G|^k} \sum_{\mathbf{g} \in G^k} \left\| f(x_\mathbf{g}; \Theta_{\mathcal{A}_t})^{(+)} \right\|^2 + \mathcal{L}(\Theta_{\mathcal{A}_t})^{(\times)},$$

where $\mathcal{U}^1(\Theta_{\mathcal{A}_t}) = \sum_{i=1}^N \mathcal{U}^1(\theta_i)$ is the cumulated initial utility function of the $N$ neurons, and

$$\mathcal{L}(\Theta_{\mathcal{A}_t})^{(\times)} = \frac{1}{2|G|^k} \sum_{\mathbf{g} \in G^k} \left\| x_{g_1 \cdots g_k} - f(x_\mathbf{g}; \Theta_{\mathcal{A}_t})^{(\times)} \right\|^2 \tag{54}$$

denotes the loss of the sigma-pi-sigma term. We know that:

$$\mathcal{U}^1(\theta_i) = \frac{k!}{C^k} \mathrm{Re}\left\langle s_w^i S_k^i \cdots S_1^i, \widehat{x}[\rho_*] \right\rangle_{\rho_*}, \tag{55}$$

where $C$ is a coefficient which equals to 1 if $\rho_*$ is real, and to 2 otherwise.

Motivated by the above loss decomposition, we now focus on (the loss of) the sigma-pi-sigma term. Specifically, we prove the following bound, which will enable us to solve the cost minimization problem.

**Theorem B.5.** *We have the following lower bound:*

$$\mathcal{L}(\Theta_{\mathcal{A}_t})^{(\times)} \geq \frac{1}{2}\left( \|x\|^2 - \frac{C\|\widehat{x}[\rho_*]\|_{\rho_*}^2}{2|G|} \right). \tag{56}$$

*The above is an equality if, and only if, the following conditions hold:*

- *For indices $\alpha_0, \beta_0, \ldots, \alpha_k, \beta_k \in \{1, \ldots, n_{\rho_*}\}$,*

$$\sum_{i=1}^N s_w^i[\alpha_0, \beta_0] \prod_{h=1}^k S_{k-h+1}^i[\alpha_h, \beta_h] = \begin{cases} \frac{C^{k+1}}{|G|n_{\rho_*}^k k!} \widehat{x}[\rho_*][\alpha_0, \beta_k] & \text{if } \beta_h = \alpha_{h+1} \text{ for } h = 0, \ldots, k-1, \\ 0 & \text{otherwise.} \end{cases} \tag{57}$$

- *If $\rho_*$ is not real, for all proper subsets $A \subset \{1, \ldots, k\}$,*

$$\sum_{i=1}^N s_w^i \otimes \bigotimes_{h \in A} S_h^i \bigotimes_{h \notin A} \overline{S_h^i} = 0. \tag{58}$$

*Proof.* From (52) and the analogous expression $\langle w^i, w^j \rangle = \frac{|G|}{C} \mathrm{Re}\left\langle s_w^i, s_w^j \right\rangle_{\rho_*}$, it follows that:

$$\frac{1}{2|G|^k} \sum_{\mathbf{g} \in G^k} \left\| f(x_\mathbf{g}; \Theta_{\mathcal{A}_t})^{(\times)} \right\|^2 = \frac{(k!)^2}{2C|G|^{k-1}} \sum_{\mathbf{g} \in G^k} \sum_{i,j=1}^N \mathrm{Re}\left\langle s_w^i, s_w^j \right\rangle_{\rho_*} \prod_{h=1}^k \mathrm{Re}\left\langle \rho_*(g_h)^\dagger, S_h^i \right\rangle_{\rho_*} \mathrm{Re}\left\langle \rho_*(g_h)^\dagger, S_h^j \right\rangle_{\rho_*}$$

$$= \frac{(k!)^2}{2C|G|^{k-1}} \sum_{i,j=1}^N \mathrm{Re}\left\langle s_w^i, s_w^j \right\rangle_{\rho_*} \prod_{h=1}^k \sum_{g \in G} \mathrm{Re}\left\langle \rho_*(g)^\dagger, S_h^i \right\rangle_{\rho_*} \mathrm{Re}\left\langle \rho_*(g)^\dagger, S_h^j \right\rangle_{\rho_*}. \tag{59}$$

By using the Schur orthogonality relations (30) and the fact that for two complex numbers $\alpha, \beta \in \mathbb{C}$ it holds that $2\mathrm{Re}\,\alpha\,\mathrm{Re}\,\beta = \mathrm{Re}\,\alpha\beta + \mathrm{Re}\,\alpha\overline{\beta}$, we deduce that:

$$\sum_{g \in G} \mathrm{Re}\left\langle \rho_*(g)^\dagger, S_h^i \right\rangle_{\rho_*} \mathrm{Re}\left\langle \rho_*(g)^\dagger, S_h^j \right\rangle_{\rho_*} = \frac{|G|}{C}\mathrm{Re}\left\langle S_h^i, S_h^j \right\rangle_{\rho_*}. \tag{60}$$

By iteratively using the same fact on real parts of complex numbers, (59) reduces to:

$$\frac{(k!)^2|G|}{2C^{k+1}} \sum_{i,j=1}^N \mathrm{Re}\left\langle s_w^i, s_w^j \right\rangle_{\rho_*} \prod_{h=1}^k \mathrm{Re}\left\langle S_h^i, S_h^j \right\rangle_{\rho_*}$$

$$=\frac{(k!)^2|G|}{(2C)^{k+1}} \sum_{i,j=1}^N \mathrm{Re}\left( \left\langle s_w^i, s_w^j \right\rangle_{\rho_*} \sum_{A \subseteq \{1,\ldots,k\}} \prod_{h \in A} \left\langle S_h^i, S_h^j \right\rangle_{\rho_*} \prod_{h \notin A} \left\langle S_h^j, S_h^i \right\rangle_{\rho_*} \right). \tag{61}$$

$$=\frac{(k!)^2|G|}{(2C)^{k+1}} \sum_{A \subseteq \{1,\ldots,k\}} \left\| \sum_{i=1}^N s_w^i \otimes \bigotimes_{h \in A} S_h^i \bigotimes_{h \notin A} \overline{S_h^i} \right\|^2_{\rho_*^{\otimes(k+1)}}.$$

When $\rho_*$ is real, all the terms in the sum above coincide (and $C = 1$). Otherwise, we isolate the term indexed by $A = \{1,\ldots,k\}$. In any case, we obtain the lower bound:

$$\frac{1}{2|G|^k} \sum_{\mathbf{g} \in G^k} \left\| f(x_{\mathbf{g}}; \Theta_{\mathcal{A}_t})^{(\times)} \right\|^2 \geq \underbrace{\frac{(k!)^2|G|}{2C^{2k+1}}}_{:=K} \left\| \sum_{i=1}^N s_w^i \otimes \bigotimes_{h=1}^k S_h^i \right\|^2_{\rho_*^{\otimes(k+1)}}$$

$$=K n_{\rho_*}^{k+1} \sum_{\alpha_0, \beta_0, \ldots, \alpha_k, \beta_k} \left| \sum_{i=1}^N s_w^i[\alpha_0, \beta_0] \prod_{h=1}^k S_{k-h+1}^i[\alpha_h, \beta_h] \right|^2. \tag{62}$$

The above bound is exact if, and only if, (58) holds. On the other hand,

$$\mathcal{U}^1(\Theta_i) = \frac{k!}{C^k} \sum_{i=1}^N \mathrm{Re}\left\langle s_w^i S_k^i \cdots S_1^i, \widehat{x}[\rho_*] \right\rangle_{\rho_*}$$

$$= \frac{k! n_{\rho_*}}{C^k} \sum_{\alpha_0, \ldots, \alpha_{k+1}} \mathrm{Re}\left( \overline{\widehat{x}[\rho_*]}[\alpha_0, \alpha_{k+1}] \sum_{i=1}^N s_w^i[\alpha_0, \alpha_1] \prod_{h=1}^k S_{k-h+1}^i[\alpha_h, \alpha_{h+1}] \right). \tag{63}$$

Each index of the outer sum of (63) corresponds to an index in the outer sum of the last expression in (62) with $\beta_h = \alpha_{h+1}$ for $h = 0, \ldots, k$. Consequently, we can lower bound (62) with a sum over these indices. This bound is exact if, and only if, the second case of (57) holds. Now, for each such index, by completing the square (in the sense of complex numbers), we obtain:

$$K n_{\rho_*}^{k+1} \left| \sum_{i=1}^N s_w^i[\alpha_0, \alpha_1] \prod_{h=1}^k S_{k-h+1}^i[\alpha_h, \alpha_{h+1}] \right|^2 - \frac{k! n_{\rho_*}}{C^k}\mathrm{Re}\left( \overline{\widehat{x}[\rho_*]}[\alpha_0, \alpha_{k+1}] \sum_{i=1}^N s_w^i[\alpha_0, \alpha_1] \prod_{h=1}^k S_{k-h+1}^i[\alpha_h, \alpha_{h+1}] \right)$$

$$= \left| K^{\frac{1}{2}} n_{\rho_*}^{\frac{k+1}{2}} \sum_{i=1}^N s_w^i[\alpha_0, \alpha_1] \prod_{h=1}^k S_{k-h+1}^i[\alpha_h, \alpha_{h+1}] - \frac{C^{\frac{1}{2}}\widehat{x}[\rho_*][\alpha_0, \alpha_{k+1}]}{(2|G|n_{\rho_*}^{k-1})^{\frac{1}{2}}} \right|^2 - \frac{C\,|\widehat{x}[\rho_*][\alpha_0, \alpha_{k+1}]|^2}{2|G|n_{\rho_*}^{k-1}}$$

$$\geq -\frac{C\,|\widehat{x}[\rho_*][\alpha_0, \alpha_{k+1}]|^2}{2|G|n_{\rho_*}^{k-1}}. \tag{64}$$

The above bound is exact if, and only if, the first case of (57) holds. This provides the desired upper bound:

$$
\mathcal{L}(\Theta_{\mathcal{A}_t})^{(\times)} - \frac{\|x\|^2}{2} = \frac{1}{2|G|^k} \sum_{\mathbf{g} \in G^k} \left\| f(x_{\mathbf{g}}; \Theta_{\mathcal{A}_t})^{(\times)} \right\|^2 - \mathcal{U}^1(\Theta)
$$

$$
\geq -\frac{C}{2|G| n_{\rho_*}^{k-1}} \sum_{\alpha_0, \dots, \alpha_{k+1}} |\widehat{x}[\rho_*][\alpha_0, \alpha_{k+1}]|^2 \tag{65}
$$

$$
= -\frac{C}{2|G|} \|\widehat{x}[\rho_*]\|_{\rho_*}^2 .
$$

$\square$

## B.3. Constructing Solutions

We now construct solutions to the cost minimization problem (still in the $\rho_*$-aligned subspace). As argued in the previous section, the sigma-pi-sigma term $f(x_{\mathbf{g}}; \Theta_{\mathcal{A}_t})^{(\times)}$ plays a special role. We will show that it is possible to construct solutions such that the remaining term $f(x_{\mathbf{g}}; \Theta_{\mathcal{A}_t})^{(+)}$ vanishes, i.e., the MLP reduces to a sigma-pi-sigma network. To this end, we provide the following decomposition of the square-free monomial $z_1 \cdots z_k$.

**Lemma B.6.** *The square-free monomial admits the decomposition*

$$
z_1 \cdots z_k = \frac{1}{k! \, 2^k} \sum_{\varepsilon \in \{\pm 1\}^k} \left( \prod_{i=1}^k \varepsilon_i \right) \sigma \left( \sum_{i=1}^k \varepsilon_i z_i \right) . \tag{66}
$$

*Proof.* After expanding the right-hand side of (66), the coefficient of the monomial $z_1^{m_1} \cdots z_k^{m_k}$ is, up to multiplicative scalar,

$$
\sum_{\varepsilon \in \{\pm 1\}^k} \left( \prod_{i=1}^k \varepsilon_i \right) \prod_{i=1}^k \varepsilon_i^{m_i} = \prod_{i=1}^k \left( 1 + (-1)^{m_i+1} \right) . \tag{67}
$$

For each $i$,

$$
1 + (-1)^{m_i+1} = \begin{cases} 0, & \text{if } m_i \text{ is even,} \\ 2, & \text{if } m_i \text{ is odd.} \end{cases} \tag{68}
$$

Hence the product is nonzero if and only if each $m_i$ is odd. Since $\sum_i m_i \leq k$, if each $m_i$ is odd then $m_1 = \cdots = m_k = 1$. Thus, the only surviving monomial is $z_1 \cdots z_k$. Note that the multiplicative constant on the right-hand side of (66) is chosen so that this monomial appears with no coefficient. $\square$

*Remark* B.7. When $\sigma(z) = z^k$, (17) is an instance of a *Waring decomposition* of the square-free monomial, i.e., an expression of $z_1 \cdots z_k$ as a sum of $k$-th powers of linear forms in the variables $z_1, \dots, z_k$. In this case, since the summands for $\varepsilon$ and $-\varepsilon$ coincide, one may choose any subset $S \subset \{\pm 1\}^k$ containing exactly one element from each pair $\{\varepsilon, -\varepsilon\}$, so that $|S| = 2^{k-1}$, and obtain the equivalent half-sum form

$$
z_1 \cdots z_k = \frac{1}{k! \, 2^{k-1}} \sum_{\varepsilon \in S} \left( \prod_{i=1}^k \varepsilon_i \right) \left( \sum_{i=1}^k \varepsilon_i z_i \right)^k . \tag{69}
$$

Even further, the square-free monomial can not be written as a sum of less than $2^{k-1}$ powers of linear forms (Carlini et al., 2011). This implies that a width of $\Omega(2^k)$ is necessary in order to implement sigma-pi-sigma units.

We are now ready to construct solutions.

**Lemma B.8.** *The following holds:*

1. *For $N \geq (k+1)n_{\rho_*}^{k+1}$ neurons, there exists $s_j^i$ and $s_w^i$ such that (57) and (58) hold.*

2. *For $N \geq (k+1)2^k n_{\rho_*}^{k+1}$ neurons, there exists $s_j^i$ and $s_w^i$ such that case 1) holds, and moreover $f(x_{\mathbf{g}}; \Theta_{\mathcal{A}_t})^{(+)} = 0$ for all $\mathbf{g} \in G^k$.*

*Proof.* **Case 1.** Up to rescaling, say, $s_w^i$, we can ignore the coefficient $C^{k+1}/(|G|n_{\rho_*}^k k!)$ in (57). For indices $\alpha, \beta$, let $E_{\alpha,\beta}$ be the matrix with a 1 in the entry $(\alpha, \beta)$, and 0 elsewhere. Let $N = n_{\rho_*}^{k+1}$. We will think of the index $i$ as a $(k+1)$-uple of indices $(\alpha_0, \ldots, \alpha_k)$. Let:

$$
\begin{aligned}
s_w^{\alpha_0,\ldots,\alpha_k} &= E_{\alpha_0,\alpha_1} \\
S_{k-h+1}^{\alpha_0,\ldots,\alpha_k} &= E_{\alpha_h,\alpha_{h+1}}, \qquad h = 1, \ldots, k-1, \\
S_1^{\alpha_0,\ldots,\alpha_k} &= E_{\alpha_k,\alpha_0}\widehat{x}[\rho_*].
\end{aligned}
\tag{70}
$$

Put simply, $s_w^i$ and $S_j^i$ correspond to 'matrix multiplication tensors'. Note that since we assumed $\widehat{x}[\rho_*]$ to be invertible, the above equations can be solved in terms of $s_j^i$. This ensures that (57) holds.

We now extend this construction to additionally satisfy (58). To this end, we set $N = (k+1)n_{\rho_*}^{k+1}$, and replicate the previous construction $k+1$ times. For an index $i$ belonging to the $j$-th copy, with $1 \le j \le k+1$, we multiply $S_h^i$ by the unitary scalar $e^{\pi \mathrm{i} j/(k+1)}$, and similarly multiply $s_w^i$ by $e^{-\pi \mathrm{i} jk/(k+1)}/(k+1)$. (When $\rho_*$ is real, we multiply by the real parts of these expressions, since in that case $s_i^j$ and $s_w$ are constrained to be real matrices.) Then each expression (58) gets rescaled by:

$$
\frac{1}{k+1}\sum_{j=1}^{k+1} e^{-\frac{2\pi \mathrm{i} j}{k+1}(k-|A|)}.
\tag{71}
$$

Since $A$ is a proper subset of $\{1, \ldots, k\}$, we have $0 < k - |A| \le k$, and thus $k - |A| \ne 0 \pmod{k+1}$. This implies that (71) vanishes, as desired.

**Case 2.** Lemma B.6 immediately implies that $2^k$ neurons can implement a sigma-pi-sigma neuron. From Case 1, we know that $(k+1)n_{\rho_*}^{k+1}$ sigma-pi-sigma neurons can solve cost minimization, which immediately implies Case 2.

$\square$

From the decomposition of the loss (53) it follows that, when the number $N$ of newly-activated neurons is large enough, Lemma B.8 describes all the global minimizers of the loss (in the space of $\rho_*$-aligned neurons $\Theta_{\mathcal{A}_t}$). Finally, we describe the function learned by such minimizing neurons, completing the proof by induction.

**Lemma B.9.** *Suppose that $N \ge (k+1)2^k n_{\rho_*}^{k+1}$ and that $\Theta_{\mathcal{A}_t}$ minimizes the loss. Then for $\mathbf{g}, p \in G^{k+1}$:*

$$
f(x_{\mathbf{g}}; \Theta_{\mathcal{A}_t})[p] = \frac{C}{|G|}\,\mathrm{Re}\,\big\langle \rho_*(g_1 \cdots g_k p)^\dagger, \widehat{x}[\rho_*]\big\rangle_{\rho_*}.
\tag{72}
$$

*Proof.* From the previous results, we know that:

$$
f(x_{\mathbf{g}}; \Theta_{\mathcal{A}_t})[p] = f(x_{\mathbf{g}}; \Theta_{\mathcal{A}_t})^{(\times)}[p] = k!\sum_{i=1}^N \mathrm{Re}\,\big\langle \rho_*(p)^\dagger, s_w^i\big\rangle_{\rho_*} \prod_{h=1}^k \mathrm{Re}\,\big\langle \rho_*(g_h)^\dagger, S_h^i\big\rangle_{\rho_*}.
\tag{73}
$$

Via computations similar to the proof of Theorem B.5, and by using (57) and (58), we deduce that the above expression equals:

$$
\begin{aligned}
&\frac{C}{|G|}\sum_{\alpha_0,\ldots,\alpha_{k+1}} \mathrm{Re}\left(\widehat{x}[\rho_*][\alpha_0,\alpha_{k+1}]\,\rho_*(p)[\alpha_0,\alpha_1]\prod_{h=1}^k \rho_*(g_{k-h+1})[\alpha_h,\alpha_{h+1}]\right) \\
&= \frac{C}{|G|}\,\mathrm{Re}\,\big\langle \rho_*(g_1 \cdots g_k p)^\dagger, \widehat{x}[\rho_*]\big\rangle_{\rho_*}.
\end{aligned}
\tag{74}
$$

$\square$

## B.4. Example: Cyclic Groups

To build intuition around the results from the previous sections, here we specialize the discussion to the cyclic group. Let $G = C_p = \mathbb{Z}/p\mathbb{Z}$ for some positive integer $p$. In this case, the group composition task amounts to modular addition. For $k = 2$, this task has long served as a testbed for understanding learning dynamics and feature emergence in neural networks (Power et al., 2022; Nanda et al., 2023; Gromov, 2023; Morwani et al., 2023).

As mentioned in Section 3.1, the irreps of $C_p$ are one-dimensional, i.e. $n_\rho = 1$ for all $\rho \in \mathcal{I}(G)$, and take form $\rho_k(g) = e^{2\pi i g k/p}$ for $k \in \{0, \ldots, p-1\}$. The resulting Fourier transform is the classical DFT. For simplicity, we assume that $p$ is odd. This will avoid dealing with the Nyquist frequency $k = p/2$, for which the following expressions are similar, but less concise.

In this case, the function learned by the network after $t - 1$ iterations of AGF (cf. (33)) takes form:

$$f(x_{\mathbf{g}}; \Theta_{\mathcal{A}})[h] = \frac{1}{p} \sum_{\rho_k \in \mathcal{I}^{t-1}} |\widehat{x}[\rho_k]| \, \cos\left(2\pi \frac{k}{p}(g_1 + \cdots + g_k + h) + \lambda_k\right), \tag{75}$$

where $\lambda_k$ is the phase of $\widehat{x}[\rho_k] = |\widehat{x}[\rho_k]| \, e^{i\lambda_k}$. After utility maximization, each neuron will take the form of a discrete cosine wave (cf. (41)):

$$
\begin{aligned}
u_j^i[g] &= A_{i,j} \cos\left(2\pi \frac{k_*}{p} g + \lambda_{i,j}\right), \\
w^i[g] &= A_{i,w} \cos\left(2\pi \frac{k_*}{p} g + \lambda_{i,w}\right),
\end{aligned}
\tag{76}
$$

where $A_{i,j}$, $A_{i,w}$ are some amplitudes, and $\lambda_{i,j}$, $\lambda_{i,w}$ are some phases, that are optimized during the cost minimization phase.

For $k = 2$, the results in the previous sections were obtained in this form, for cyclic groups, by Kunin et al. (2025). Our results therefore extend theirs to arbitrary groups and to arbitrary sequence lengths $k$.

## C. Discussion on Assumptions

In this appendix, we discuss two assumptions underlying our analysis in Section 4: the alignment assumption used in the cost-minimization phase, and the use of polynomial activation functions. For each assumption, we provide additional evidence beyond the main text, combining empirical diagnostics with theoretical arguments where possible.

### C.1. Alignment During Cost Minimization (Assumption 4.2)

Assumption 4.2 states that neurons activated during utility maximization remain aligned with the selected irrep $\rho_*$ during the subsequent cost-minimization phase. This assumption is needed to stitch together the selection rule from utility maximization into a full irrep-by-irrep learning trajectory. Here, we provide both empirical and theoretical evidence that this behavior is natural in the regime studied in the main text.

First, we empirically track the fraction of each neuron's Fourier energy contained in its dominant irrep throughout training; see Figure 6. This metric provides a direct diagnostic of whether individual neurons remain spectrally concentrated during cost minimization, rather than only measuring macroscopic quantities such as the loss. As the initialization scale decreases, the behavior becomes increasingly stepwise: neurons rapidly concentrate on a single dominant irrep, remain approximately concentrated during the subsequent cost-minimization phase, and then new neurons activate at later stages. This supports the view that Assumption 4.2 holds approximately in practice, especially in the small-initialization regime where the AGF approximation is most accurate.

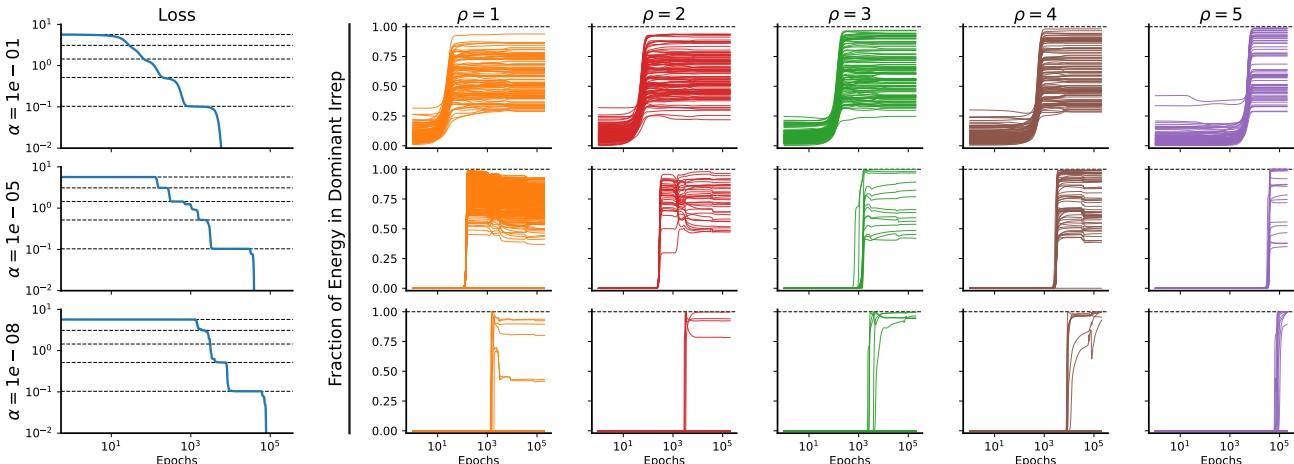

*Figure 6.* **Empirical evidence for alignment during cost minimization.** We track the fraction of each neuron's Fourier energy contained in its dominant irrep throughout training. As the initialization scale decreases (rows from top to bottom), neurons become increasingly spectrally concentrated: they rapidly align with a single irrep when activated and remain approximately aligned during the subsequent cost-minimization phase. This provides empirical evidence that Assumption 4.2 holds approximately in the small-initialization regime studied in our theory.

Second, we provide theoretical evidence for the assumption by establishing it for sigma-pi-sigma networks, which are central in our theory. More specifically, in the notation of Section B.2, we show that the derivatives of the loss $\mathcal{L}(\Theta_{\mathcal{A}_t})^{(\times)}$ of the sigma-pi-sigma term in directions aligned to irreps other than $\rho_*$ (and its conjugate) vanish.

**Lemma C.1.** *Consider a vector $v \in \mathbb{R}^{|G|}$ of the form $v[g] = \mathrm{Re} \left\langle \rho(g)^\dagger, s_v \right\rangle_\rho$ for some $\rho \neq \rho_*, \overline{\rho_*}$ and $s_v \in \mathbb{C}^{n_\rho \times n_\rho}$. Then:*

$$\frac{\partial \mathcal{L}^{(\times)}}{\partial v}(\Theta_{\mathcal{A}_t}) = 0. \tag{77}$$

*Proof.* A direct computation yields:

$$\left\langle v, \frac{\partial \mathcal{L}^{(\times)}}{\partial u_j^i}(\Theta_{\mathcal{A}_t}) \right\rangle = \frac{1}{|G|^k} \sum_{\mathbf{g} \in G^k} \sigma' \left( \sum_{h=1}^k \left\langle u_h^i, x_{g_h} \right\rangle \right) \left\langle v, x_{g_j} \right\rangle \left\langle w^i, f(x_{\mathbf{g}}; \Theta_{\mathcal{A}_t})^{(\times)} - x_{g_1 \cdots g_k} \right\rangle. \tag{78}$$

Note that $\sigma'$ is a polynomial of degree $k-1$. Thus, by expanding $\sigma'((\sum_{h=1}^k \left\langle u_h^i, x_{g_h} \right\rangle)) \left\langle v, x_{g_j} \right\rangle$ similarly to the proof of Lemma B.2, the only monomial where all the group elements $g_1, \ldots, g_k$ appear is $\left\langle u_h^i, x_{g_1} \right\rangle \cdots \left\langle u_h^i, x_{g_k} \right\rangle \left\langle v, x_{g_j} \right\rangle$. Any other monomial, when multiplied by $\left\langle w^i, f(x_{\mathbf{g}}; \Theta_{\mathcal{A}_t})^{(\times)} \right\rangle$ or by $\left\langle w^i, x_{g_1 \cdots g_k} \right\rangle$, will vanish. But by Schur orthogonality (30), the surviving monomial also vanishes once multiplied, since $\rho \neq \rho_*, \overline{\rho_*}$. A similar argument holds for the derivative with respect to $w^i$, concluding the proof. $\square$

As a consequence, if we considered the dynamics for the sigma-pi-sigma network alone—or, equivalently, if $f(x_{\mathbf{g}}; \Theta_{\mathcal{A}_t})^{(+)}$ vanished throughout training—then Assumption 4.2 would hold. However, this is not the case, in general. Yet, the above result still shows, at least, that the sigma-pi-sigma loss component $\mathcal{L}^{(\times)}$ does not contribute to steering the network away from the $\rho_*$-aligned subspace. Due to the loss decomposition (53), the only component contributing to this is the squared norm of $f(x_{\mathbf{g}}; \Theta_{\mathcal{A}_t})^{(+)}$.

### C.2. Polynomial Activation Functions

Our theoretical analysis uses polynomial activation functions because they expose the $\Sigma$-$\Pi$ term that drives our analysis of utility maximization and cost minimization. This assumption is less restrictive than it may seem: many smooth activation functions are well approximated near the origin by polynomial Taylor truncations. We therefore expect the mechanism described in the main text to persist whenever the activation has a non-negligible degree-$k$ component and the irrep gap is large enough relative to the approximation error.

To test this, we train networks with several standard activations on the cyclic group task; see Figure 7. The results are broadly consistent with the theory. Linear networks fail to learn, as predicted by Lemma 3.5. ReLU, Tanh for $k = 3$, and GeLU for $k = 2$ display learning curves similar to the polynomial setting, including a comparable order of irrep learning. By contrast, Tanh for $k = 2$ and GeLU for $k = 3$ behave differently, appearing to learn several irreps more simultaneously. We do not view this as a contradiction, but rather as evidence that these activations fall outside the specific mechanism analyzed here: their Taylor expansions at zero lack the relevant degree-$k$ monomial. In these cases, optimization must rely on a different strategy, which we leave as an interesting direction for future work.

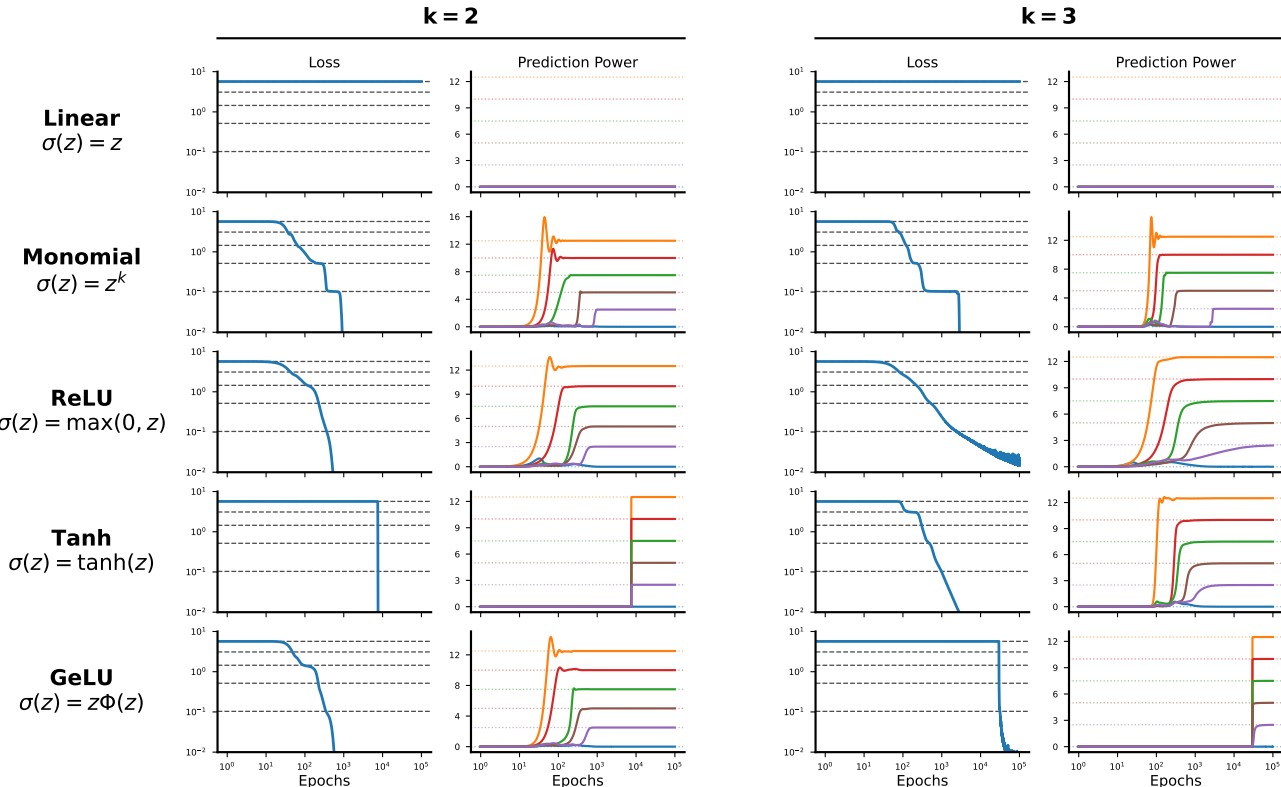

*Figure 7.* **Effect of the activation function.** We train networks on the cyclic group composition task using several standard activation functions. Linear networks fail to learn, while ReLU, Tanh for $k = 3$, and GeLU for $k = 2$ exhibit learning dynamics similar to the polynomial setting. Tanh for $k = 2$ and GeLU for $k = 3$ behave differently, consistent with the fact that their Taylor expansions at zero lack the relevant degree-$k$ monomial driving the mechanism analyzed in Section 4.

# D. Experimental Details

Below we provide experimental details for the figures in this work. Code used to produce these figures is publicly available at github.com/geometric-intelligence/sequential-group-composition.

## D.1. Constructing a Datasets for Sequential Group Composition

We provide a concrete walkthrough of how we construct the datasets used in our experiments, specifically the experiments used to produce Figure 3.

1. **Fix a group and an ordering.** Let $G = \{g_1, \ldots, g_{|G|}\}$ be a finite group with a fixed ordering of its elements. This ordering defines the coordinate system of $\mathbb{R}^{|G|}$ and the indexing of all matrices below; any other choice yields an equivalent dataset up to a global permutation of coordinates.

2. **Regular representation.** For each $g \in G$, define its left regular representation $\lambda(g) \in \mathbb{R}^{|G| \times |G|}$ by $\lambda(g)e_h = e_{gh}$ for all $h \in G$, where $\{e_h\}$ is the standard basis of $\mathbb{R}^{|G|}$. Equivalently, $\lambda(g)_{i,j} = 1$ if $gg_j = g_i$ and 0 otherwise. These matrices implement group multiplication as coordinate permutations.

3. **Choose an encoding template.** Fix a base vector $x \in \mathbb{R}^{|G|}$ satisfying the mean-centering condition $\langle x, \mathbf{1} \rangle = 0$, which removes the trivial irrep component. In many experiments, we construct $x$ in the group Fourier domain by specifying matrix-valued coefficients $\widehat{x}[\rho] \in \mathbb{C}^{n_\rho \times n_\rho}$ for each $\rho \in \mathcal{I}(G)$ and applying the inverse group Fourier transform $x = F\widehat{x}$.

   For higher-dimensional irreps ($n_\rho > 1$), we typically use scalar multiples of the identity, $\widehat{x}[\rho] = \alpha_\rho I$, which are full-rank and empirically yield stable learning dynamics. To induce clear sequential feature acquisition, we choose the diagonal values $\alpha_\rho$ using the following heuristics:

   - **Separated powers.** Irreps with similar power tend to be learned simultaneously; spacing their magnitudes produces distinct plateaus.
   - **Low-dimensional dominance.** Clean staircases emerge more reliably when lower-dimensional irreps have substantially larger power than higher-dimensional ones. This is related to the dimensional bias we verrify in Appendix D.2.
   - **Avoid vanishing modes.** Coefficients that are too small may not be learned and fail to produce a plateau.

4. **Generate inputs and targets.** The encoding of each group element is given by its orbit under the regular representation, $x_g := \lambda(g)x$. For a sequence $\mathbf{g} = (g_1, \ldots, g_k)$, the network input is the concatenation $x_{\mathbf{g}} = (x_{g_1}, \ldots, x_{g_k}) \in \mathbb{R}^{k|G|}$ and the target is $y_{\mathbf{g}} = x_{g_1 \cdots g_k} \in \mathbb{R}^{|G|}$. The full dataset consists of all $|G|^k$ pairs $(x_{g_1}, \ldots, x_{g_k}) \mapsto x_{g_1 \cdots g_k}$ for $(g_1, \ldots, g_k) \in G^k$.

## D.2. Empirical Verification of Irrep Acquisition

We now empirically test the theoretical ordering predicted by Equation (13) by constructing controlled encodings in which the score of each irrep can be independently tuned. This allows us to directly observe how the predicted bias toward lower-dimensional representations emerges and strengthens with sequence length.

We consider the sequential group composition task with the Dihedral group $D_3$ and a mean-centered one-hot encoding for $k = 2, 3, 4, 5$. For $k = 2$, we use a learning rate of $5.0 \times 10^{-5}$ and an initialization scale of $2.00 \times 10^{-7}$. As $k$ increases to 3, 4, and 5, the learning rate is held constant at $1.0 \times 10^{-4}$ while the initialization scale is increased from $5.0 \times 10^{-5}$ to $5.0 \times 10^{-4}$ and finally $2.0 \times 10^{-3}$. As we can see in the following experiment shown in Figure 8, the time between learning the one-dimensional sign irrep (brown) and the two-dimensional rotation irrep (blue) increases as the sequence length $k$ gets larger, confirming our prediction based on the theory.

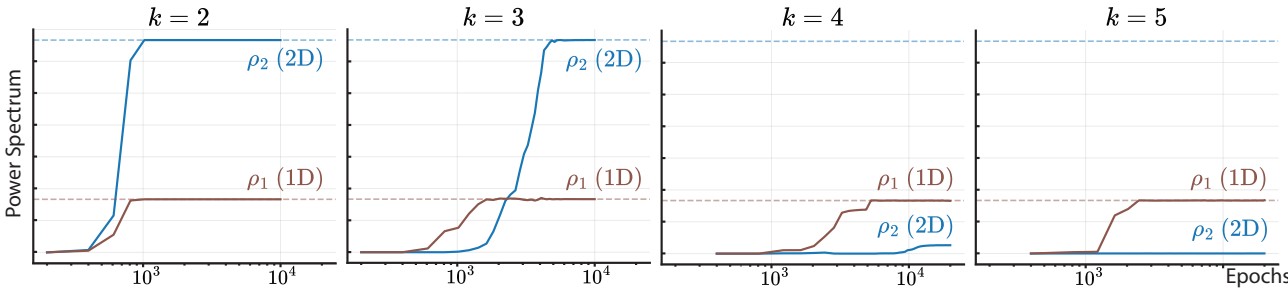

*Figure 8.* **Verifying dimensional bias in $D_3$.** Power Spectrum components $\rho_1$ (1D) and $\rho_2$ (2D) during training across sequence lengths $k = 2, 3, 4, 5$ for the group $D_3$. The bias towards learning low-dimensional irreps first increases with $k$.

## D.3. Scaling Experiments: Hidden Dimension, Group Size, and Sequence Length

Figure 4 is generated by training a large suite of two-layer networks on sequential group composition for cyclic groups $G = C_p$. Across all experiments we use a mean-centered one-hot encoding and consider sequence lengths $k = 2, 3, 4, 5, 6$. For each value of $k$, we perform a grid sweep over both the group size and the hidden dimension. Specifically, we vary the group size as $|G| = 2, 3, 4, \ldots, 22$ (21 values) and the hidden dimension as $H = 18, 36, 54, \ldots, 378$ (21 values), yielding a total of 2205 trained models.

**Normalized loss.** Because the initial mean-squared error scales inversely with the group size, we report performance using a normalized loss. For a mean-centered one-hot target, the squared target norm is approximately constant, while the MSE

averages over $|G|$ output coordinates, giving an initial loss $\mathcal{L}_{\text{init}} \approx 1/|G|$. We therefore define the normalized loss as

$$\mathcal{L}_{\text{norm}} = \frac{\mathcal{L}_{\text{final}}}{\mathcal{L}_{\text{init}}}, \tag{79}$$

which allows results to be compared directly across different group sizes.

**Training setup.** All models are trained online, sampling fresh sequences at each optimization step. We use the Adam optimizer with learning rate $10^{-3}$, $\beta_1 = 0.9$, and $\beta_2 = 0.999$, and a batch size of $1{,}000$ samples per step. Gradients are clipped at a norm of $0.1$ for stability. Weights are initialized as

$$W_{\text{in}} \sim \mathcal{N}\left(0, \frac{\sigma_k^2}{k\,|G|}\right), \qquad W_{\text{out}} \sim \mathcal{N}\left(0, \frac{\sigma_k^2}{H}\right), \tag{80}$$

where the initialization scale is adjusted per sequence length as $\sigma_k = \sigma_{\text{base}}^{3/(k+1)}$ with $\sigma_{\text{base}} = 0.01$. This gives $\sigma_2 = 0.01$, $\sigma_3 \approx 0.032$, $\sigma_4 \approx 0.063$, and $\sigma_5 = 0.1$. Training is stopped early once a $99.9\%$ reduction in loss is achieved, i.e., when $\mathcal{L}_{\text{final}} < 10^{-3}\,\mathcal{L}_{\text{init}}$, or after a maximum of $10^6$ optimization steps.

**Theory boundary.** To interpret the empirical phase diagrams, we overlay the sufficient-width threshold predicted by our construction in Equation (18). In the cyclic case $G = C_n$, this expression simplifies for three reasons. First, all irreps of $C_n$ are one-dimensional, so $n_\rho = 1$. Second, $\mathcal{J}(G)$ indexes non-trivial irreps modulo conjugation, giving $|\mathcal{J}(C_n)| = \lfloor |G|/2 \rfloor$. Third, for the monomial activation $\sigma(z) = z^k$, the Waring decomposition in Equation (17) has only $2^{k-1}$ distinct terms, since the terms indexed by $\varepsilon$ and $-\varepsilon$ coincide. Thus, for the cyclic experiments in Figure 4, the sufficient-width threshold becomes $H \geq (k+1)2^{k-1}\lfloor |G|/2 \rfloor$, which is the boundary shown in the phase diagrams.

# E. Technical Remarks on Some Related Work

As discussed in Section 2, the group composition task has been previously considered in the literature. Here, we summarize the group-theoretical analyses from the literature, and discuss how they relate to our work.

Gromov (2023) first constructed DFT-based solutions for the binary modular addition task (i.e., $k = 2$ and $G = C_p$ is the cyclic group) via shallow two-layer networks with quadratic activation ($\sigma(z) = z^2$). Similar solutions were shown to be optimal in terms of margin minimization by Morwani et al. (2023). The latter work also considered more general finite groups $G$, assuming they only admit real irreps. Stander et al. (2023) provided an alternative construction, exploiting cosets of subgroups instead of irreps. The irrep-based and coset-based formalisms have been unified by Wu et al. (2025). Interestingly, there is a subtle connection between the latter and our analysis. The fact that in our construction the matrices $s_j^i$ have rank 1 (see (70)) is, essentially, equivalent to the main claim by Wu et al. (2025) that neurons implement so-called $\rho$-sets. Formally, $s_j^i$ corresponds, in their notation, to $\mathbf{a} \otimes \mathbf{b}$—an arbitrary rank-1 matrix as $\mathbf{a}$ and $\mathbf{b}$ vary. Lastly, these constructions were extended to the sequential version of the modular addition task ($k > 2$) by Li et al. (2024). In this case, our width bound of $H = \mathcal{O}(k2^k)$ improves theirs of $\mathcal{O}(4^k)$ (our exponential factor $2^k$ is, in a sense, optimal—see Remark B.7).

We remark that our work not only generalizes the above settings to the sequential task for arbitrary finite groups and for (two-layer) networks with arbitrary polynomial activations, but also provides a dynamical analysis of how irreps emerge during training (in the vanishing initialization regime, via the AGF framework). Concurrently to our work, a similar setting has been considered by He et al. (2026), who studies dynamics and feature emergence of modular addition, in the same regime and via tools similar to AGF. Their analysis only considers a single phase, i.e., the first learned frequency (the 'winning lottery ticket'). Our results can be viewed as an extension to a full multi-phase setting and to arbitrary groups.

