# OpenReview forum: "Sequential Group Composition: A Window into the Mechanics of Deep Learning"
_ICML.cc/2026/Conference — ICML 2026 regular_

### Official Review · Reviewer_4dcN · 2026-03-09

**Soundness:** 4
**Presentation:** 3
**Significance:** 4
**Originality:** 4
**Overall Recommendation:** 5
**Confidence:** 3

**Summary:**

Paper presents an interesting take on how deep learning works when data has some compositionality via studying how neural networks learn sequential composition of group elements. Their choice of algebraic structures for this task allows them to develop a relevant mathematical framework and demonstrate that two-layer networks learn to do it one irreducible representation at a time and in order determined by Fourier statistics of the encoding. They further show that efficient solutions possibly exploiting associativity exist within deeper architectures. This framework and its results on studying how neural networks learn this compositional task is illuminating.

**Compliance With Llm Reviewing Policy:**

Affirmed.

**Final Justification:**

The paper presents an interesting study on how neural networks learn sequential composition of group elements. The authors develop a mathematical framework and provide sufficient experimental validation for their claims. Although code was not provided, the methodology was sufficiently clear that this reviewer successfully replicated the findings on additional groups not examined in the paper, confirming the framework's utility for addressing questions in the theory of deep learning. The rebuttal addressed my concerns and reinforced my assessment. My confidence score 3 acknowledges the impossibility of fully verifying serious math within one month of reviewing, so it would otherwise be higher.

**Key Questions For Authors:**

Q1: Can you provide evidence beyond macroscopic loss curves that neurons remain aligned during cost minimization? For instance, tracking the fraction of each neuron's Fourier energy concentrated on its dominant irrep throughout training.

Q2: Do you have training experiments for the RNN (Eq. 19) and multilayer MLP (Eq. 22) architectures? If so, do the learned weights actually resemble the constructions in Equations (21) and (24-25), or does gradient descent find qualitatively different solutions?

Q3: All experiments train on the full dataset of |G|^k sequences. How does performance degrade with partial training data? Does the irrep-by-irrep ordering persist, and do lower-dimensional irreps generalize from fewer samples as the theory might suggest?

Q4: Is the 2^k factor in Equation (18) tight, or is it an artifact of the specific Waring decomposition used? Are there lower bounds on the width needed?

Q5: Section 5.3 suggests solvable groups admit constant-depth transformer solutions. Do you have any preliminary evidence — theoretical or empirical — for this, or is it purely speculative at this stage?

**Limitations:**

yes

**Strengths And Weaknesses:**

**Strengths**

The presentation is clean and the authors attempt to outline complex math in a digestible manner. The work is sound (some caveats below), original and advances our understanding of machine learning.

**Weaknesses**

**W1**
The abstract claims the stepwise irrep-by-irrep learning dynamics are proved, but this overstates what the analysis actually delivers. The inductive argument (Section 4.2) requires Assumption 4.2 — that newly activated neurons remain aligned to a single irrep throughout cost minimization — and this assumption is justified only by appeal to empirical consistency, not by any property of the gradient flow. Since cost minimization operates outside the slow manifold governed by AGF, there is no formal reason alignment should be preserved, and the decoupling via Schur orthogonality (Eq. 50), while suggestive, is never shown to imply dynamical stability of the aligned subspace. What is proved unconditionally is the selection rule (Theorem 4.1): the Fourier statistics of the encoding determine which irrep is learned next. But closing the induction — and hence the full "one irrep at a time" claim — depends on an uncontrolled assumption. The paper would benefit from either (a) direct empirical verification that individual neuron weights remain spectrally concentrated during cost minimization, or (b) a theoretical argument, even heuristic, for why the gradient dynamics preserve alignment.

**W2**
Brief primer on harmonic analysis over groups tries to introduce everything the reader must know to understand the paper, however, makes a gap at least once. When talking about representations, authors state that they are focusing on unitary matrices and unitary representations. The significance of this choice is never explained. It is never mentioned that this does not lead to a loss of generality, as every representation of a finite group is equivalent to a unitary one. In fact, the entire harmonic analysis machinery — Plancherel theorem (Eq. 29), Schur orthogonality (Eq. 30), the convolution theorem (Eq. 28), the unitary change of basis F — all rely on the representations being unitary and real encoding needs it implicitly as well.

**W3**
The abstract says "we show how deeper models exploit the associativity of the task to dramatically improve this scaling." From my point of view, I see that deeper models can exploit associativity, but the fact that deeper models do exploit associativity when trained is not shown. It is possible I've overlooked something and it would be nice to be directed where to look.

---

> ### Author Rebuttal · Authors · 2026-03-30
>
> We really appreciate your comprehensive review, and your suggested improvements. We will respond to the weaknesses and questions raised.
>
> **W1.** This is an important question, and we agree this is a point of our paper we could strengthen. In the updated version, we will provide both empirically theoretically evidence for assumption 4.2 in a new appendix section:
>
> - Empirically, we have implemented your suggested experiment, that provides evidence that the assumption is satisfied, approximately, in practice – see our answer to Q1 below.
>
> - Theoretically, we will add a proof that the assumption holds for sigma-pi-sigma networks, which are central in our analysis – see our reply to **Q1 of Rev. iEFe** for details. While this does not provide a proof of the assumption for the networks we study, it gives an indication that it is theoretically natural, and actually shows that there is a significant gradient component pushing the MLP to the irrep-aligned subspace.
>
> **W2.** We agree that the role of unitarity should be clarified better. As you mentioned, this is not limiting: every finite-dimensional representation is equivalent to a unitary one, and the representation $\lambda$ used in our construction is itself unitary. We use the unitary formulation because the Fourier transform and its properties are most naturally stated in that setting. We will add a sentence to Sec. 3.1 making this “without loss of generality” point explicit.
>
> **W3.** You are correct. We will rephrase this sentence as "...deeper models **can** exploit the associativity of the task to...". Additionally, we will include empirical evidence for this – see our answer to Q2 below.
>
> **Q1.** This is a great suggestion. We think the correct metric is the Fourier energy concentrated on the dominant irrep, normalized by the maximum energy it eventually achieves. Showing this metric through training provides useful empirical evidence for assumption 4.2, as it postulates that neurons that eventually activate will start with $0.0$, suddenly jump to $1.0$, and then stay at $1.0$. Please see the file `assumption.pdf` in [this anonymized repository](http://tinyurl.com/icml-group-comp) for a new figure we will add to our appendix, showing the fraction of energy in dominant irrep for the cyclic group $C_{11}$ as a function of initialization scale. It can be seen that, as the initialization scale decreases, the behavior becomes more stepwise, as assumption 4.2 suggests.  We are working on adding a third row to Fig. 2 showing this energy metric for each neuron across all the groups we study.
>
> **Q2.** As suggested, we include additional experiments to complement this section, demonstrating that indeed GD finds solutions similar to the associativity-based solution we construct. Please see our reply to **Q2 of Rev. iEFe** for a discussion on a new experiment (file `rnn.pdf` in the above repository) showing how a GD-trained RNN self-organizes to have a block-diagonal mixing matrix. We are currently working on a similar experiment for the multilayer MLP.
>
> **Q3.** We believe that a characterization of how performance degrades with partial training data, despite interesting, falls beyond the scope of this paper. We expect that the training will depend on the fraction of data used, and on how the fraction is chosen (e.g., uniformly or adversarially). We also wish to point out that while Fig. 1 is trained on all $|G|^k$ samples, Fig. 3 and 5 were both run online, sampling continuously from the input space. Even for moderately sized $|G|$ and $k$, this is the only practical way to train these networks. We will add a discussion in the conclusion, highlighting that a complete exploration of how our analysis changes with partial data is an important and interesting direction for future work.
>
> **Q4.** The $2^k$ factor is actually optimal. Indeed, we show that, to minimize the loss, the MLP needs to implement sigma-pi-sigma units. In other words, the network needs to implement some Waring decomposition. This requires $\Omega(2^k)$ neurons, since the square-free monomial (Eq. 17) can not be written as a sum of a smaller number of  polynomial $k$-th powers. This is a known fact from polynomial algebra, termed “Waring rank” – see  Carlini et al., The Solution to Waring’s Problem for Monomials. We will highlight this in the updated version. Also see `updatedfigure3.png` in the above repository for an extended version of Fig. 3, demonstrating our bound empirically as $k$ increases.
>
> **Q5.** Our claim that solvable groups admit constant-depth transformer solutions is mostly speculative, and is based on the connection between groups and finite-state automata. We believe that this result can be deduced from Thm. 2 by Liu et al., but the details need to be established. We hope to work on extensions of our analysis to FSAs and transformers.
>
> Thank you again for your feedback! We hope we addressed your concerns and that you continue to advocate for our paper.

---

> > ### Author Rebuttal · Reviewer_4dcN · 2026-04-03
> >
> > My questions have been addressed, which confirms that the score I chose is correct. At the same time, the score is positive, reflecting my overall favorable opinion of the work. In order to increase my score, I would need to see how the final version of the paper with all revisions turns out, as I believe incorporating all those changes constitutes a significant revision.

---

> > > ### Author Response · Authors · 2026-04-03
> > >
> > > Thank you again for your thoughtful review — we are very glad that our rebuttal fully resolved your concerns, and we sincerely appreciate your positive score and overall favorable assessment of the paper.

---

### Official Review · Reviewer_goVV · 2026-03-10

**Soundness:** 3
**Presentation:** 4
**Significance:** 2
**Originality:** 2
**Overall Recommendation:** 4
**Confidence:** 4

**Summary:**

This paper introduces the sequential group composition task as a tractable setting for analyzing how neural networks learn structured sequence computations. Given a sequence of encoded finite-group elements, the goal is to predict the encoding of their product. The paper combines group representation theory, Fourier analysis over groups, and the alternating gradient flow (AGF) framework to analyze two-layer polynomial networks in the vanishing-initialization regime. The main result is that shallow networks learn one irreducible representation at a time in an order determined by the Fourier statistics of the encoding, but exact learning requires width exponential in sequence length. The paper then gives constructive arguments showing that recurrent and deeper feedforward architectures can exploit associativity to compute the same task far more efficiently.

**Compliance With Llm Reviewing Policy:**

Affirmed.

**Final Justification:**

The authors addressed several of my main concerns by discussing scope and robustness more carefully, and improving the framing of the deeper-model claims. Important limitations still remain, especially the narrow analytical regime and the more constructive nature of the deep-model results. Even so, I find the paper sound, clear, and original as a theory contribution. The final version should make the scope boundaries especially explicit so readers do not overgeneralize the strongest results. I retain a positive (4) final view of this paper.

**Key Questions For Authors:**

- How much of the shallow-network learning-order result survives for more standard activations and initialization scales?
- How sensitive are the results to the choice of encoding vector x beyond the generic orbit-based assumptions used here?

**Limitations:**

The paper would benefit from a more explicit limitations discussion. In particular, it should state more directly that the main results depend on a highly stylized task, polynomial activations, and vanishing-initialization theory, and that the deeper-model sections are more constructive than optimization-theoretic.

**Strengths And Weaknesses:**

Strengths:
- The theoretical setup is coherent: the task, the harmonic-analysis framework, and the AGF learning analysis fit together well.
- The paper proves both a meaningful nonlinear lower point (linear models cannot solve the task under the stated assumptions) and a shallow-vs-depth efficiency gap.
- The paper is well organized for a theory paper, and Figure 4 is a useful aid for readers less familiar with representation theory.
- The connection between irrep learning order, encoding statistics, and architectural efficiency is elegant and distinctive.

Weaknesses:
- The most important results rely on a narrow regime: polynomial activations and vanishing initialization for shallow networks.
- The deeper-model results are mostly constructive rather than full gradient-based learning analyses.
- The practical scope is limited because the setting is highly stylized and the main theorems depend on specific analytical assumptions.
- The transformer implications are not yet demonstrated at the same level as the shallow-network theory.

---

> ### Author Rebuttal · Authors · 2026-03-30
>
> **Q: Other activations**
> This is a great question. We use polynomial activations because they make the theory clean and tractable, since they expose the $\Sigma$-$\Pi$ term driving our analysis. This is less restrictive than it may seem: many smooth activations are approximated by polynomial (Taylor) truncations. We expect our theory to apply when the activation has an degree-$k$ approximation, and the irrep gap (Eq. 13) is large enough w.r.t. the approximation error.
>
> Inspired by your question, we performed an experiment with different activations (linear, ReLU, Tanh, GeLU) for the cyclic group – see the file `nonlinearity.pdf` in [this anonymized repository](http://tinyurl.com/icml-group-comp). The results are consistent with our theory:
>
> - Linear networks can’t learn, as our Lemma 3.5 establishes.
>
> - ReLU, Tanh for $k=3$, and GeLU for $k=2$, all display similar learning curves (i.e. order of irreps learned) to the setting we study.
>
> - Tanh for $k=2$ and GeLU for $k=3$ are peculiar: they appear to learn all irreps simultaneously. We do not view this as a contradiction, but rather as evidence that these activations fall outside the specific mechanism we analyze: their Taylor expansions at $0$ lack the required degree-$k$ monomial ($z^2$ and $z^3$ respectively). In these cases, optimization must therefore rely on a different strategy, which we view as an interesting direction for future work.
>
> We will include this result and discussion in the updated version.
>
> **Q: Larger initialization**
> The vanishing initialization regime is an artificial setting, where feature learning phenomena are known to emerge clearly. Specifically, AGF enables a mathematical theory of feature learning. Yet, similarly to above, if the initialization is close enough to $0$ (as often happens in practice), then the approximation by AGF is suitable. See the file `assumption.pdf` in the above repository for a new figure (which we will add to the appendix), showing how, as initialization scale decreases, the alignment to our theory improves.
>
>
>
> **Q: Encoding vector**
> We are not sure we follow your question. The statistics of the encoding vector $x$ determines the order of irreps learned (Eq. 13). Our theory applies to $x$ satisfying the assumptions from Sec. 4.2. These are satisfied by almost all $x$, and are actually not fundamental. The mean-centering is needed only because we consider bias-less MLPs, the non-degeneracy assumption is only used in Eq. 70 (which can be solved under milder but more convoluted conditions), and the last condition is only needed to ensure irreps are learned separately (the proofs carry over without it). Thus, we see these conditions as very mild.
>
>
> **W: Scope and Practicality**
> In our view, the purpose of this paper is to provide a lens into the mechanics of deep learning, via a formal and comprehensive study of a specific task (group composition). Even further, we discuss in Sec. 6 how this task is a special case of finite-state automata and regular languages, hinting at connections with NLP. While we agree on the “toy” nature of this task, we argue that it captures some essential ingredients of more general and realistic settings. Beyond the task, a similar perspective applies to our choice of activation function and initialization regime. As we argued above, these choices enable a natural and clean mathematical treatment, and elucidate what happens in practical scenarios.
>
> **W: Depth**
> It is correct that we are unable to analyze deep networks dynamically. Mathematically, this is extremely challenging. In fact, the AGF framework does not apply, in its current form, to more than two layers – see our reply to **Q2 of Rev. fdZR** for more details on this. It is common in the literature that shallow networks are analyzed dynamically, while deeper networks are analyzed via their expressivity. Yet, we agree that our arguments for deep networks could be strengthened. To this end, we have performed new experiments to demonstrate that the construction we give is found by GD, and emerges through training – see our reply to **Q2 of Rev. iEFe**.
>
> **W: Transformers**
> Understanding how a Transformer learns sequential group composition is, indeed, a very interesting problem. In Sec. 5.3, we briefly speculate that Transformers might admit constant-depth solutions for solvable groups. However, we believe that a deeper exploration of this idea would constitute a whole work on its own, and consider it an important future direction.
>
> **W: Limitations**
> We completely agree that our work would benefit from a thorough limitation section. We will add it, and incorporate the points discussed above.
>
> Thank you again for your constructive feedback. We hope we addressed your questions and concerns. If we have, we would appreciate it if you could raise your score to reflect this.

---

> > ### Author Rebuttal · Reviewer_goVV · 2026-04-02
> >
> > I agree that extending the current model to a broader scope and deeper networks/Transformers is challenging. Since my initial score is already positive, I will maintain my current score.

---

> > > ### Author Response · Authors · 2026-04-02
> > >
> > > We thank you for the follow-up, and we are glad to hear that your assessment of the paper remains positive.

---

### Official Review · Reviewer_iEFe · 2026-03-12

**Soundness:** 3
**Presentation:** 3
**Significance:** 3
**Originality:** 3
**Overall Recommendation:** 4
**Confidence:** 3

**Summary:**

This paper studies how neural networks learn structured operations from sequences using the task of sequential group composition, where the model predicts the cumulative product of a sequence of group elements. The authors show that when inputs are mean-centered, the task cannot be solved by linear models, making nonlinear representation learning essential.

The paper provides a theoretical analysis of two-layer neural networks using the Alternating Gradient Flow framework. It shows that networks learn irreducible representations (irreps) of the underlying group sequentially during training, and that the order of learning depends on the Fourier statistics of the input encoding. Lower-dimensional irreps tend to be learned first, especially for longer sequences.

The authors also analyze the architectural efficiency of different models. They show that a two-layer MLP can solve the task but requires exponential width in the sequence length, whereas deeper architectures can exploit associativity: RNNs compute the result in $k$ steps and deep MLPs in $O(\log k)$ layers. The paper argues that this task serves as a useful testbed for studying representation learning dynamics and the benefits of depth.

**Compliance With Llm Reviewing Policy:**

Affirmed.

**Final Justification:**

The paper presents an original and mathematically interesting analysis of representation learning dynamics on sequential group composition, with a clear connection between group structure, learned irreducible representations, and the benefits of depth. I find the work strong in originality and reasonably significant as a theory-driven contribution, and the presentation is generally clear despite the technical density.

The rebuttal was helpful and addressed several of my concerns, especially by clarifying the role of Assumption 4.2 and by adding supporting empirical evidence for the deep-model discussion. However, my main reservations remain: some core arguments still rely on restrictive assumptions, and the results for deeper architectures are still more representational/constructive than fully dynamical. Taking both the paper and the rebuttal into account, I believe the strengths and weaknesses remain broadly balanced as in my original review, so I am maintaining my initial score.

**Key Questions For Authors:**

1. The theoretical analysis relies on an assumption that newly activated neurons remain aligned with the same irrep during cost minimization. Under what conditions can this assumption be formally justified?

2. The results for deeper architectures mainly show representational efficiency. Do gradient-based training methods reliably discover these efficient solutions in practice?

3. The learning order of irreps depends on the Fourier statistics of the input encoding. How robust is this phenomenon under more realistic or learned embeddings?

4. How sensitive are the exponential width lower bounds for two-layer networks to the choice of activation function or architectural modifications?

**Limitations:**

The paper does not explicitly discuss limitations or potential societal impacts.

**Strengths And Weaknesses:**

Strengths

The paper proposes a clean and well-motivated theoretical setting for studying representation learning dynamics. By framing the task in terms of group composition and Fourier analysis, the authors obtain interpretable results about how neural networks learn structured features.

A key strength is the theoretical result showing that two-layer networks learn group irreducible representations sequentially during training. This provides a principled explanation of feature learning dynamics that extends previous observations from modular arithmetic tasks.

Another important contribution is the analysis of architectural efficiency. The paper clearly demonstrates that shallow networks require exponential width to solve the task, while deeper models such as RNNs or deep MLPs can compute the composition much more efficiently by exploiting associativity. This offers a clear and intuitive explanation of the benefits of depth.

The experiments, although limited, support the main theoretical claims. In particular, they show that lower-dimensional irreps are learned earlier and that shallow networks exhibit the predicted width limitations.

Weaknesses

The theoretical analysis relies on restrictive assumptions. In particular, the key results depend on the vanishing initialization regime and on an assumption that newly activated neurons remain aligned with the same irrep during training. While the authors argue that this behavior is empirically observed, its general validity is not fully justified.

In addition, the analysis of deeper architectures mainly shows existence of efficient representations, rather than demonstrating that gradient-based training reliably finds them. As a result, the learning dynamics of deep models are not studied as rigorously as those of the two-layer networks.

Finally, the empirical evaluation is relatively limited and focuses on small finite groups. While appropriate for a theory-driven paper, additional experiments demonstrating the behavior in broader settings would strengthen the paper’s claims about the general relevance of the task.

---

> ### Author Rebuttal · Authors · 2026-03-30
>
> We thank you for your review, and are happy to hear your appreciation of our theoretical analysis.  We will respond individually to the questions raised:
>
> **Q1**. We agree that assumption 4.2 is critical. The same assumption is required in the original work on AGF (Kunin et al.). While we are not able to fully prove this assumption in the generality we consider, we wish to highlight two points:
>
> - We can prove that this assumption *holds for sigma-pi-sigma* networks (which are central in our work – see Sec. 4.2). More formally: the irrep-aligned subspace is invariant to the gradient flow of these networks. This is relevant, since the loss of the MLP contains the loss of a sigma-pi-sigma network as an additive component (see Eq. 53). While this does not imply the assumption for the whole MLP, it at least shows that the gradient of the loss contains a force dragging the parameters to the irrep-aligned subspace. We believe that this force is dominant in the dynamics of the MLP. This is informally suggested by our overall theory; as argued in lines 277-284, all the computations are determined by the sigma-pi-sigma term, and everything else is extraneous and gets cancelled out. We believe that this motivates the assumption convincingly, from a theoretical perspective. In the updated version, we will include this proof and discussion.
>
> - To complement the theory, we will also include an *empirical analysis* showing that this assumption is approximately satisfied in practice at small initialization scales. Please see our reply to **Q1 for Rev. 4dcN** for more details.
>
> **Q2**. We acknowledge that our section on deep networks is only concerned with representational efficiency, and does not analyze the learning dynamics. This is due to the theoretical intractability; while the gradient dynamics of shallow networks is feasible to study with AGF, deeper networks are not. This said, to strengthen this section, we will include new empirical results. We have reverse-engineered the weights of a GD-trained RNN, showing that it converges to our constructed solution – see the file `rnn.pdf` in [this anonymized repository](http://tinyurl.com/icml-group-comp).  It can be seen that the RNN self-organizes to a block-diagonal mixing matrix $W_{\text{mix}}$, as described in Sec. 5.1. We also see the RNN undergoes similar stepwise dynamics, specializing neurons of $W_{\text{out}}$ to individual irreps. We are currently working on a similar experiment for the deep MLP.
>
> **Q3**. We believe that considering encodings beyond our orbit-based one is an interesting question. However, it is challenging to answer in general, since, in principle, the group can be encoded via an arbitrary map $\gamma: G \rightarrow \mathbb{R}^p$ (where $p = |G|$ is the input dimension). We do not see any reason to expect, in general, that the order of the learned irreps will be governed by Fourier statistics, since $\gamma$ might have no connection with harmonic analysis (e.g., it could even be non-linear). Yet, under the reasonable assumptions that the encoding map $\gamma$ is linear and $G$-equivariant (i.e., preserves the group structure), $\gamma$ is necessarily an orbit-based encoding (for some encoding vector $x$), and our theory applies. From this perspective, orbit-based encodings are rather general. We will remark on this in the updated version.
>
>
> **Q4**. With respect to the activation function, our exponential bound applies rather generally. Our results hold for arbitrary polynomial activations (of degree $\leq k$). This is a large class, since polynomials can approximate arbitrary continuous functions (by Weierstrass Approximation). Thus, our results might even generalize beyond polynomials via approximation arguments. However, this would require a careful extension of our argument based on the Waring decomposition (which is where the exponential bound comes from). Regarding architectural modifications, we do not see any obstruction in generalizing our proof to small nuances, such as skip connections. Indeed, for skip connections, the Waring-based argument will still provide an exponential bound. On the other hand, drastic architectural changes can significantly affect the bound. An example is already discussed in the paper; for sigma-pi-sigma networks, the bound improves to linear in $k$ (see point 1 in Lemma B.8). While these networks are uncommon in practice, this showcases how the bound depends drastically on the architecture. See our reply to **Q4 of Rev. 4dcN** and **Q1 of Rev. goVV** for a discussion on optimality of our bound and influence of activation function, respectively.
>
> Thank you again for your constructive feedback. We hope we addressed your questions and concerns. If we have, we would appreciate it if you could raise your score to reflect this.

---

> > ### Author Rebuttal · Reviewer_iEFe · 2026-04-01
> >
> > The rebuttal addresses several key concerns (especially Assumption 4.2 and deeper models) with additional justification and planned experiments, which improves confidence in the claims. In particular, the invariance result for sigma-pi-sigma networks and empirical evidence for alignment make the assumption more plausible.
> >
> > However, some concerns remain only partially resolved. The core limitation --- lack of a full proof of the assumption for general MLPs --- still stands, and the results for deep networks remain largely empirical or constructive rather than theoretically grounded. These are important but not fatal issues, and I would have a few follow-up questions regarding generality and robustness.

---

> > > ### Author Response · Authors · 2026-04-02
> > >
> > > Thank you for your follow-up. We are happy to hear that our rebuttal resolved several of your main concerns, and that the additional theoretical justification for Assumption 4.2, together with the new empirical results, strengthened your assessment of our paper.
> > >
> > > We agree that these are important points. In fact, we believe that a complete dynamical analysis of deep networks would be a substantial project on its own, and see it as a major direction for future work. In this work, we focused on analyzing the shallow-network dynamics, and complementing this analysis with a non-dynamical discussion on deep networks.
> > >
> > > We would be happy to address any follow-up questions. Unfortunately, we are not sure the platform allows it. As we understand, no further interaction between authors and reviewers is possible. If your major concerns have been addressed, we would appreciate it if you could consider raising your score to reflect this.

---

### Official Review · Reviewer_fdZR · 2026-03-19

**Soundness:** 2
**Presentation:** 2
**Significance:** 2
**Originality:** 2
**Overall Recommendation:** 4
**Confidence:** 4

**Summary:**

This paper attempts to provide a theoretical explanation for what neural networks learn in order to solve group composition tasks of arbitrary length k.
In this general setting, they use the Alternating Gradient Flow (AGF) framework to show that one hidden-layer networks learn irreducible representations, one at a time. Moreover, such networks require 2^k neurons to solve the task.
For deeper networks, they do not provide an AGF proof, but they provide constructions for what RNNs and MLPs learn.

**Compliance With Llm Reviewing Policy:**

Affirmed.

**Final Justification:**

On the condition that the authors remove section 5 as outlined in their most recent rebuttal, I believe that this paper should be accepted on the merits of their methodology for proving their AGF result in the group task setting.

**Key Questions For Authors:**

- Since the prior work on training neural networks on group multiplications always uses cross entropy loss, do you think your results using mean squared error will match what networks learn when trained with a different loss function, being cross entropy loss.
- Why are recurrent and multilayer networks outside the AGF framework? Are there obvious obstructions to applying it?
- Can the AGF framework be extended to non-group settings?
- For the results in section 5 "Benefits of Depth: Leveraging Associativity", have you reverse engineered networks that you trained on these problems and do the trained networks match your construction? What experiments could you run to show your construction matches reality? Similarly, can you show RNNs learn your construction?

**Limitations:**

See questions above.

**Strengths And Weaknesses:**

**Strengths**

This paper addresses an important question in interpretability research, being that the community needs interpretable toy models, where we understand what the solutions learned by deep neural networks (of multiple architectures) are. This paper steps toward making a contribution in this area by presenting a study of what neural networks of various architectures learn in the toy setting of group multiplications of arbitrary length, which is an interesting setting the community is currently focused on. Indeed, there are many papers that study group multiplications, so the scope of this paper is strong, and relevant. Overall, I like what this paper is trying to do, and believe the work has potential. The proof about gradient flow for arbitrary groups seems novel, though limited to 1 hidden layer settings, I think this is still a contribution to the field.

**Weaknesses**


A significant drawback of this paper is that it doesn't cite, and therefore doesn't clearly interface with many results in the literature. This is my largest concern with this work, and leads me to believe it possibly needs a rewrite to explain how the work fits into the narrative established in the literature.

- "Towards a unified and verified understanding of group-operation networks" Wu et al. ICLR 2025.
- "Fourier Circuits in Neural Networks and Transformers: A Case Study of Modular Arithmetic with Multiple Inputs" Li et al. 2025.
- "Uncovering a universal abstract algorithm for modular addition in neural networks" McCracken et al. NeurIPS 2025.
- "Can Neural Networks Learn Small Algebraic Worlds? An Investigation Into the Group-theoretic Structures Learned By Narrow Models Trained To Predict Group Operations" Kvinge et al. TAG-DS 2025.
- "On the geometry and topology of representations: the manifolds of modular addition" Moisescu-Pareja et al. ICLR 2026.
- "On the Mechanism and Dynamics of Modular Addition: Fourier Features, Lottery Ticket, and Grokking" He et al. 2026.

Wu et al. is a necessary citation as it improves on Stander et al.'s results and is more rigorous in that their reverse engineering is backed by theory and formally verified.
Li et al., says that modular addition of length k (the cyclic group multiplication of length k) requires 2^k neurons. This is a necessary prerequisite citation as this work is aiming to generalize this to any group, not just cyclic groups.
McCracken et al. proved that O(log(n)) irreducible irreps are learned (frequencies) by deep networks, and the frequencies carve the dataset into cosets, and multiple neurons with the same frequency parameterize a Cayley graph. They prove that deep neural networks (MLPs + transformers) both learn the same algorithm to solve cyclic group multiplication, essentially learning the divide and conquer strategy of the Chinese Remainder Theorem (which intersects a logarithmic number of cosets to acquire the correct answer).
Kvinge et al. train networks on group tasks and interpret what algebraic properties they learn, e.g. commutativity, the identity, etc.
Moisescu-Pareja et al. proved that the neural representations learned by deep neural networks on cyclic group multiplication were tori or discs, giving a geometric and topological interpretation that unified  multiple different claims of what could be learned by networks trained on cyclic group multiplication. He et al. provides a gradient flow/ODE analysis of the learning dynamics for cyclic group multiplication.

One of your citations, Stander et al., makes a suggestion for caution: future work on group multiplications should go above and beyond to ensure theoretical models match empirical reality. This suggestion was made as their paper found prior work that was widely disseminated had presented an incorrect model of what neural networks were learning.

Indeed, this commentary from Stander et al. is where my weaknesses are centered. This paper makes strong claims of what deep neural networks are learning, but does not present sufficient empirical evidence to make these claims concrete.

While the associativity construction is nice, I am not convinced that it's what deep multi-layer networks learn to solve a group multiplication of length k. I think this construction must be provided alongside empirical evidence that shows it's what networks learn. This means deep neural networks must be reverse engineered, and the construction must be found directly, or at least be inferrable from empirical measurements. Additionally, once this is done, would this construction interface well with the interpretations provided by Wu et al., Stander et al., McCracken et al., Moisescu-Pareja et al., etc, or would it provide a new, competing account; since this work seeks to generalize group multiplications, it must fit nicely with results on simpler example tasks.

To summarize: I like the work but need to see two things. 1) convincing empirical results backing the claims that are made. 2) Explanations of how the results in this paper generalize prior work, since this work aims to generalize these prior works.

---

> ### Author Rebuttal · Authors · 2026-03-30
>
> We thank you for your review, and are glad to hear that you recognize the scope and relevance of our work. We will address the weaknesses and questions raised.
>
> **Related works**. We apologize for missing the works you mentioned, which we agree are relevant. In the revised version, we will include a discussion on them, summarized here:
>
> - Wu et al. unifies the “irrep perspective” with the “coset perspective” by Stander et al. Interestingly, there is a subtle connection to our analysis. The matrix $s_j$  (defining the linear combination of $\rho$) has rank 1 (Eq. 70). This is, essentially, equivalent to the main claim in Wu et al. that the neurons implement so-called $\rho$-sets. Formally, $s_j$ corresponds to $\bf{a} \otimes \bf{b} $ (in their notation) – an arbitrary rank-1 matrix as $\bf{a}$ and $\bf{b}$ vary. Thus, our theory recovers, to an extent, their results.
>
> - Li et al. is closely related, since it also considers multiple inputs ($k>2$). Our results can be seen as a (dynamical) generalization of theirs to arbitrary groups. Our width bound $\mathcal{O}(k 2^k)$ improves theirs $\mathcal{O}(4^k)$. The exponential $2^k$ is actually optimal – see our reply to **Q4 of Rev. 4dcN**.
>
> - The construction by McCracken et al. fits into Sec. 5.3, where we briefly discuss how algebraic decompositions can be exploited for more efficient solutions (e.g., with $\log(|G|)$ depth, akin to Liu et al.). In fact, the Chinese Remainder Thm. can be seen as an instance of a group decomposition (specifically, a Jordal-Hölder filtration).
>
> We highlight that the main distinction from the works above is that our analysis (for shallow nets) is *dynamical*. Our goal is to describe how irreps emerge during training. This is possible thanks to three original ingredients: non-Abelian groups, giving rise to a rich space of irreps; orbit-based input encodings, breaking task symmetries; and the vanishing-initialization regime, making AGF applicable. All this allows us to describe the mechanisms and the order of irrep emergence. In this respect, He et al. is perhaps the closest work to ours in spirit: it also studies modular addition dynamically, in the same regime, using tools similar to AGF (which they cite). Our results can be viewed as extending that perspective to a multi-phase setting and to arbitrary groups. Please note that He et al. appeared on arXiv after the ICML submission deadline. We will cite it, alongside the other works you mentioned.
>
>
>
> **Empirics**. While we see our work as primarily theoretical, we agree with the cautionary point raised, and that more empirical results would strengthen our paper. To this end, we have implemented new experiments, which we will include in paper. The results can be found at [this anonymized repository](http://tinyurl.com/icml-group-comp). Please see our replies to other reviewers, where we discuss these results:
>
>
> - Testing Assumption 4.2 (reply to **Rev. 4dcN**)
>
>
> - Different nonlinearities (reply to **Rev. goVV**)
>
>
> - Extended Fig. 3 (reply to **Rev. 4dcN**)
>
>
> - RNN experiment (reply to **Rev. iEFe**)
>
> We highlight the last result, where we have reverse-engineered the weights of an RNN, showing that it learns our construction (see Q4 below).
>
> **Questions**.
> Here we will address the questions you raised:
>
> **Q1**. Some parts of our analysis may extend to other losses, including cross-entropy. This is especially plausible for the utility maximization step, which depends only weakly on the specific loss. The cost minimization step is more loss-sensitive and would require separate arguments.
>
>
> **Q2**.  AGF applies only to two-layer networks because the theory relies on the network output decomposing as a sum over neurons, as in Eq. 6. This structure is not directly available in deeper or recurrent networks. That said, extensions of AGF may still be possible, and Kunin et al. discuss several directions. Developing such an extension would be a substantial project in its own right and lies beyond our present scope.
>
>
> **Q3**.  The AGF framework itself is not limited to group settings: it applies to two-layer networks trained on arbitrary tasks. Beyond that, we believe our specific analysis may extend to algebraic settings more general than groups, such as semi-automata and formal grammars, as discussed in Sec. 6.
>
>
> **Q4**.  As mentioned, we have performed experiments showing an RNN learns our construction – see the file `rnn.pdf` in the above repository, and our reply to **Q2 of Rev. iEFe**. The RNN is convenient for reverse-engineering, since the specific structure of its mixing matrix $W_{\text{mix}}$ (see Sec. 5.1) makes it detectable. We are working on the corresponding visualization for a deep MLP.
>
> We appreciate the time and care you put in the review, which helped us improve our work. We hope our response has addressed your concerns. If that’s the case, we would be grateful if you would reconsider your current score.

---

> > ### Author Rebuttal · Reviewer_fdZR · 2026-04-04
> >
> > Thank you for your response, but this rebuttal does not address my core concerns.
> >
> > I am unconvinced that multi-layer networks learn the associativity construction and find this very concerning because this subfield has had papers published prematurely before, where claims were later falsified. As I said in my initial review, I like the part of this paper dedicated to dynamics. I would have been willing to give this paper a higher score were this paper just sticking to what it does well: the dynamics and theoretical study of shallow networks. I believe this part of the paper is correct, and worth publishing.
> >
> > However, I remain very unconvinced of the claims in section 5.1. The new empirical result for an RNN learning the claimed solution is insufficient. A proper analysis would be quantitative, not qualitative, and over many random seeds, and also show over different groups that RNNs and MLP networks learn the associativity construction matching the logarithmic result of the construction. To be frank, this section feels shoehorned into an otherwise good paper about dynamics that didn't need speculative constructions for multilayer networks. I will note that all other reviewers found this problematic.
> >
> > > "In addition, the analysis of deeper architectures mainly shows existence of efficient representations, rather than demonstrating that gradient-based training reliably finds them. As a result, the learning dynamics of deep models are not studied as rigorously as those of the two-layer networks." - **iEFe**
> >
> > > "The deeper-model results are mostly constructive rather than full gradient-based learning analyses." - **goVV**
> >
> > > "The abstract says "we show how deeper models exploit the associativity of the task to dramatically improve this scaling." From my point of view, I see that deeper models can exploit associativity, but the fact that deeper models do exploit associativity when trained is not shown. " - **4dcN**
> >
> > Resultingly, I recommend this paper for rejection, because its acceptance could cause harmful misunderstandings about what's being learned by networks with depth on groups. Again, I caution for a second time, that papers with similar scope (networks trained on groups) presented constructions that were mathematically correct (e.g. Chughtai et al., Nanda et al., Zhong et al.), but were not learned in practice by networks as shown by Stander et al., and Moisescu-Pareja et al.
> >
> > I would be open to accepting this paper if the depth associativity construction was completely removed. My suggestion to the authors is to publish these two papers separately. One about dynamics (which I believe the current results complete), and a second about the associativity construction, which would require significant empirical verification and would result in an impactful contribution were it to be true across all groups. If the authors commit to removing the associativity construction and focusing the paper on the dynamics, I would recommend this paper for weak acceptance. Otherwise, I am confident stating that this work is incomplete and requires substantial further work to back the deep associativity construction, either theoretically or empirically.

---

> > > ### Author Response · Authors · 2026-04-06
> > >
> > > Thank you for the follow-up and for the feedback, which we find constructive and actionable.
> > >
> > > After taking your criticism into consideration, **we are open to removing Sec. 5**. Our plan is to replace Sec. 5 with a longer version of Sec. 6 (Discussion), where we will shortly discuss the current contents of Sec. 5, phrased as speculative directions for future work (with no formalism). We will also mention, as a limitation, the difficulty of extending AGF to deeper networks, as discussed in our previous reply. This new part of Sec. 6 will be similar in spirit and tone to the current Sec. 5.3, which discusses, shortly and speculatively, how algebraic properties might be leveraged to find more efficient solutions. This will result in more available space in the paper, which we will use to 1) extend Sec. 4 with high-level summaries of proof ideas and a deeper discussion on Assumption 4.2, and 2) extend Sec. 2 by incorporating the works you mentioned in the review, as discussed in our previous reply.
> > >
> > > We wish to state that, in our opinion, the content of this section has a value. Its purpose is to contrast the inefficiency we establish for two-layer networks with an explicit demonstration that depth can, in principle, obtain much more efficient solutions. It does it by providing formal and constructive solutions, without claiming a result about dynamics in deep networks (which is extremely hard). Indeed, while we agree that all other reviewers flagged Sec. 5 as lacking empirical evidence, some of them did appreciate its contents. For example, **Rev iEFe** who mentioned it explicitly among the main strengths, and the other reviewers mentioned that their concerns in this merit were addressed by the rebuttal. That said, we agree that we should have stated its scope more clearly. We especially understand the cautionary point about constructions that, despite being natural, might not be learned by networks in practice. As also suggested by **Rev. 4dcN**, this section needs a rewording, clearly stating that the intent is constructive, and explaining its limitations, especially in terms of the cautionary tales you raised.
> > >
> > > In conclusion, we understand your constructive criticism, and thank you for it. We will remove Sec. 5 as described above, and would appreciate it if you raised your score. Still, we think a more appropriate and less extreme option is a substantial rewording. Thus, we would like the Area Chair to also weigh in on what they think the best course of action is.

---

### Decision · Program_Chairs · 2026-04-30

**Decision:**

Accept (regular)

**Comment:**

Reviewers broadly agreed that this is a sound, clear, and well-motivated theoretical contribution. In particular, the analysis of shallow networks was recognized as a solid contribution toward understanding representation learning dynamics, and in itself was viewed as potentially sufficient to warrant publication.

At the same time, reservations were raised regarding the analysis of deep networks, which formalizes a mechanistic hypothesis through a constructive argument. Reviewers noted that, while this construction makes the hypothesis plausible, the paper does not provide empirical evidence that it reflects the actual dynamical behavior of trained deep networks. Two reviewers viewed this as an acceptable limitation, whereas one reviewer indicated that their support for acceptance would depend on removing the depth-associativity analysis altogether.
Overall, there was broad agreement that the claims concerning deeper models should be framed more carefully, and that the status of this part of the analysis should be clarified, so as to avoid overclaiming or giving a misleading impression of the dynamics of trained networks. The authors’ rebuttal proposed concrete and constructive ways of addressing these concerns, and the discussion already led to useful clarification and improvement.

On balance, I recommend acceptance. In my view, the paper makes a meaningful theoretical contribution, particularly through its analysis of shallow networks, and the concerns raised about the deep-network portion are best addressed through clearer framing rather than by withholding acceptance. I strongly encourage the authors to carefully incorporate the reviewers’ feedback in the final version, especially in clarifying the interpretation and limitations of the deep-network results so as to avoid ambiguity or overstatement.